# Balanced Source Terms for Wave Generation Within The Hasselmann Equation

Vladimir Zakharov[1,2,3,4], Donald Resio[5], and Andrei Pushkarev[1,2,3,4]

[1]Department of Mathematics, University of Arizona, Tucson, AZ 85721, USA
[2]Lebedev Physical Institute RAS, Leninsky 53, Moscow 119991, Russia
[3]Novosibirsk State University, Novosibirsk, 630090, Russia
[4]Waves and Solitons LLC, 1719 W. Marlette Ave., Phoenix, AZ 85015, USA
[5]Taylor Engineering Research Institute, University of North Florida

*Correspondence to:* Andrei Pushkarev (dr.push@gmail.com)

**Abstract.**

The new ZRP wind input source term (Zakharov et al., 2012) is examined for its theoretical consistency via numerical simulation of the Hasselmann equation. The results are compared to field experimental data, collected at different sites around the world, and theoretical predictions based on self-similarity analysis. Consistent results are obtained for both limited fetch and duration limited statements.

## 1 Introduction

The scientific description of wind driven wave seas, inspired by solid state physics statistical ideas (see, for instance, Nordheim (1928)), was proposed by Hasselmann (1962, 1963) in the form of Hasselmann Equation (hearafter HE), also known as kinetic equation for waves:

$$\frac{\partial \varepsilon}{\partial t} + \frac{\partial \omega_k}{\partial \boldsymbol{k}} \frac{\partial \varepsilon}{\partial \boldsymbol{r}} = S_{nl} + S_{in} + S_{diss} \tag{1}$$

where $\varepsilon = \varepsilon(\omega_k, \theta, \boldsymbol{r}, t)$ is the wave energy spectrum, as a function of wave dispersion $\omega_k = \omega(k)$, angle $\theta$, two-dimensional real space coordinate $\boldsymbol{r} = (x, y)$ and time $t$. $S_{nl}$, $S_{in}$ and $S_{diss}$ are the nonlinear, wind input and wave-breaking dissipation source terms, respectively. Hereafter, only the deep water case, $\omega = \sqrt{gk}$ is considered, where $g$ is the gravity acceleration and $k = |\boldsymbol{k}|$ is the absolute value of the vector wavenumber $\boldsymbol{k} = (k_x, k_y)$.

Since Hasselmann's work, Eq.(1) has become the basis of operational wave forecasting models such as WAM, SWAN and Wavewatch III (Tolman, 2013; SWAN). While the physical oceanography community consents on the general applicability of Eq.(1), there is no consensus agreement on universal parameterizations of the source terms $S_{nl}$, $S_{in}$ and $S_{diss}$.

## 1.1 The $S_{nl}$ term and Weak Turbulence Theory

The HE is a specific example of the kinetic equation for quasi-particles, widely used in different areas of theoretical physics. There are standard methods for its derivation. In the considered case, two forms of $S_{nl}$ term were derived by different methods from the Euler equations for free surface incompressible potential flow of a liquid by Hasselmann (1962, 1963) and Zakharov and Filonenko (1966). Resio and Perrie (1991) showed that they are identical on the resonant surface

$$\omega_{\boldsymbol{k_1}} + \omega_{\boldsymbol{k_2}} \quad = \quad \omega_{\boldsymbol{k_3}} + \omega_{\boldsymbol{k_4}} \tag{2}$$

$$\boldsymbol{k_1} + \boldsymbol{k_2} \quad = \quad \boldsymbol{k_3} + \boldsymbol{k_4} \tag{3}$$

The $S_{nl}$ term is the complex nonlinear operator acting on $\varepsilon_k$, concealing hidden symmetries (Zakharov and Filonenko, 1967; Zakharov et al., 1992) and cubic with respect to the spectrum $\varepsilon$.

Eq.(1) sometimes is called "the Boltzmann equation" by the oceanographic community, although this is a misconception. The Boltzmann equation, derived in the nineteenth century for description of gas kinetics, is quadratic, rather then cubic, with respect to the distribution function.

To understand the relation and difference between the Boltzmann equation and HE, on should recollect above mentioned Nordheim (1928) equation. This equation, applicable to quantum quasi-particles, contains both quadratic and cubic terms. Hence, the Boltzmann equation and HE present opposite limiting cases of a general quantum kinetic equation.

Purely cubical (applicable to classical waves, not to classical particles) systems are relatively new objects in physics. Such equations describe the simplest case of the wave turbulence by the theory, which is called the "Weak Turbulence Theory" (WTT) (Zakharov et al., 1992).

It is clear now that the WTT can be used for the description of very broad class of physical phenomena, including waves in magneto-hydrodynamics (Galtier et al., 2000), waves in nonlinear optics (Yousefi, 2017), gravitational waves in the universe (Galtier and Nazarenko, 2017; de Oliveira et al., 2013), plasma waves (Balk, 2000; Yoon et al., 2016), capillary waves (Pushkarev and Zakharov, 1996; Yulin, 2017; Tran, 2017), and Kelvin waves in super-fluid Helium (L'vov and Nazarenko, 2010).

It is unfortunate that the discussion of HE in the context of WTT has been overlooked by a major part of the oceanographic community for many years now. The community accepts, nevertheless, HE as the basis for the operational wave forecasting models, therefore believing de-facto in WTT without fully appreciating its ramifications.

The WTT essentially differs from the kinetic theory of classical particles and quantum quasi-particles. In the "traditional" gas kinetics (both classical and quantum) the basic solutions are thermodynamic equilibrium spectra, such as Boltzmann and Plank distributions. In the WTT such solutions, though formally existing, play no role – they are non-physical. The physically essential solutions are the non-equilibrium Kolmogorov-Zakharov spectra (or KZ-spectra, Zakharov et al. (1992)), which are the solutions of the corresponding kinetic equation

$$S_{nl} = 0 \tag{4}$$

The simplest one is the Zakharov-Filonenko (hearafter ZF) solution (Zakharov and Filonenko, 1966), which is the sub-class of KZ solutions:

$$\varepsilon \simeq \frac{P^{1/3}}{\omega^{-4}} \tag{5}$$

where $P$ is the energy flux toward high wave numbers.

The accuracy advantage of knowing the analytical expression for the $S_{nl}$ term, also known in physical oceanography as XNL, is overshadowed by its computational complexity. Today, none of the operational wave forecasting models can afford to perform XNL computations in real time. Instead, the operational approximation, known as DIA and its derivatives, is used to replace this source term. The implication of such simplification is the inclusion of a tuning coefficient in front of nonlinear term; however, several publications have shown that DIA does not provide a good approximation of the actual XNL form. The

paradigm of replacement of the XNL by the DIA and its variations leads to even more grave consequences: other source terms must be adjusted to allow the model Eq.(1) to produce desirable results. In other words, deformations suffered by XNL model due to the replacement of $S_{nl}$ by its surrogates, need to be compensated by non-physical modification of other source terms to achieve reasonable model behavior in any specific case, leading to a loss of physical universality in HE model.

## 1.2   Operational formulations for the wind energy input $S_{in}$ and wave energy dissipation $S_{diss}$ terms

In contrast to $S_{nl}$, the knowledge of $S_{in}$ and $S_{diss}$ source terms is poor; furthermore, both include many heuristic factors and coefficients. The creation of a reliable, well justified theory of $S_{in}$ has been hindered by strong turbulent fluctuations, uncorrelated with the wave motions, in boundary layer over the sea surface. Even one of the most crucial elements of this theory, the vertical distribution of horizontal wind velocity in the region closest to the ocean surface, where wave motions strongly interact with atmospheric motions, is still the subject of debate. The history of the development of different wind input

forms is full of heuristic assumptions, which fundamentally restrict the magnitude and directional distribution of this term. As a result, the values of different wind input terms scatter by a factor of $300 - 500\%$ (Badulin et al., 2005; Pushkarev and Zakharov, 2016). For example, experimental determination of $S_{in}$, as provided by direct measurements of the momentum flux from the air to the water, cannot be rigorously performed in a laboratory due to gravity waves dispersion dependence on the water depth, as well as problems with scale effects in laboratory winds. Additional information on the detailed analysis of current state of

the art of wind input terms can be found in Pushkarev and Zakharov (2016).

     Similar to the wind input term, there is little consent on the parameterization of the source dissipation term $S_{diss}$. The physical dissipation mechanism, which most physical oceanographers agree on, is the effect of wave energy loss due to wave breaking, while there are also other dubious ad-hoc "long wave" dissipation source terms, having heuristically justified physical explanations. Currently, there is not even an agreement on the location of wave breaking events in Fourier space. The approach

currently utilized in operational wave forecasting models mostly relies on the dissipation, localized in the vicinity of the spectral energy peak. Recent numerical experiments show (Pushkarev and Zakharov, 2016; Dyachenko et al., 2015; Zakharov et al.,

2009), however, that such approach does not pass most of the tests associated with the essentially nonlinear nature of the HE Eq.(1).

## 1.3 Roadmap for the construction and verification of balanced source terms

The next chapters present a balanced set of wind energy input and wave energy dissipation source terms, based on WTT and experimental data analysis. Further, they are numerically checked to comply WTT predictions and experimental observations. As mentioned above, contrary of previous attempts for building the detailed-balance source terms, the current approach is neither based on the development of a rigorous analytic theory of turbulent atmospheric boundary layer, nor on reliable and repeatable air to ocean momentum measurements. The new $S_{in}$ is constructed in the artificial way realizing, in a sense, "the poor man approach", based on the finding of two-parameter family of HE self-similar-solutions and their restriction to the single-parameter one with the help of comparison with the data of experimental observations, accumulated for several decades.

Section 2 presents experimental evidence of wave energy spectra characteristics in the form of specific regression line, found by Resio et al. (2004a). The analytic form of this regression line will play crucial role in narrowing the circle of possible outcomes, obtained with WTT analysis.

Section 3 studies self-similar solutions of HE – kinetic equation for surface ocean waves, starting with the analysis of the behavior of the dissipationless HE in infinite space, containing the wind source term in power function form. This approach is similar in spirit to one realized by Zakharov and Filonenko (1967) for finding the solution of the equation $S_{nl} = 0$ in the infinite Fourier domain, which derived ZF spectrum $\varepsilon \sim \frac{P^{1/3}}{\omega^4}$, where $P$ is the energy flux toward high wave numbers. The Fourier domain in both situations does not contain any dissipation function: its role is played by infinite phase volume as the effective energy sink at infinitely high wave numbers.

Such a situation is similar to one realized in incompressible liquid turbulence for large Reynolds numbers, where the energy distribution is given by famous Kolmogorov spectrum, transferring the energy from large to small scales, where the energy dissipation is realized due to viscosity, but the viscosity coefficient, i.e. the dissipation details, are not included into final Kolmogorov spectrum expression. The ZF spectrum and its further KZ generalizations are in this sense the ideological Kolmogorov spectrum counterparts, having the significant difference, that the Kolmogorov spectrum is a plausible conjecture, while KZ spectra are the exact solutions of the wave kinetic equation.

Since the current research is application oriented, it is important to understand, why this formally academic approach is connected with reality. In this context, there is no such thing as the dissipation at infinitely small waves in nature: however, it is clear that the existence of an absorption at sufficiently high finite frequencies provides a wave scale in real applications that still preserves the KZ solutions, found from HE equation in infinite space.

As was mentioned before, this statement was confirmed in different physical context with radically different inertial ranges (the wave-numbers band between characteristic energy input and characteristic wave energy dissipation), showing KZ solutions with different corresponding indices. As for the considered case of gravity waves on the surface of a deep fluid, KZ spectra have been routinely observed in multiple experiments. The results, published before 1985, are summarized by Phillips (1985).

Thereafter, they were observed and discussed by Resio and Long (2007). A complete survey of all measurements requires separate comprehensive paper, which is in our plans for the future.

The assumed close relation of HE in the infinite space and finite domain, bounded by high-frequency dissipation, also has a much deeper meaning, consisting in the fact that $S_{nl}$ is the leading term of HE (Zakharov, 2010; Zakharov and Badulin, 2011).

This allows further use of the solutions found from "zero-dissipation" HE Eq.(13) in infinite space for "practical" Fourier domains with the dissipation localized at finite high enough wave-numbers. They take the form of two-parameter family of self-similar solution, which can be further restricted to the single-parameter one using experimental regression dependence, presented in the section 2. These self-similar solutions present realistic HE solutions and describe a broad class of wave energy spectra, observed in ocean and wave tanks experiments.

The indices, corresponding to self-similar solutions, allow to wrap up the section 3 with the specific form of wind input term in infinite phase space, called ZRP wind input term (Zakharov et al., 2012; Pushkarev and Zakharov, 2016) with an arbitrary coefficient in front of it. Now, the theoretical part of the wind source term $S_{in}$ construction is finished, but the obtained model is not suitable yet for numerical simulation, since to perform in finite phase space, it has to be augmented with the wave breaking dissipation term.

Section 4 explains the dissipation function, used in the presented model. The wave breaking dissipation, also known as "white-capping dissipation", is an important physical phenomenon, not properly studied yet for the reasons of mathematical and technological complexity. Longuet-Higgins (1980a, b) achieved important results, but didn't accomplish the theory completely. Irisov and Voronovich (2011) studied the wave-breaking of short waves, "squeezed" by surface currents, caused by longer waves, and showed that they become steep and unstable. Our explanation is simpler, but has the same consequences: the

"wedge" formation, preceding the wave breaking, causes the "fat tail" appearance in Fourier space. Subsequent smoothing of the tip of the wedge is equivalent to a "chopping off" of the developed high-frequency tail in Fourier space – a sort of natural low-pass filtering – leading to the loss of the wave energy. Both scenarios consequences of smoothing the wave surface, and are indirectly confirmed by the numerical experiments presented in the current study.

There is considerable freedom in choosing specific analytic form of such high-frequency dissipation term, given the lack of

a generally accepted rigorous derivation for this mechanism. Consequently, one can choose a preferred and possibly justify it, but any particular choice will be questioned since it will remain somewhat artificial. Because of that, our motivation was that at the current stage of development, we considered simplicity as a primary motivating factor. Instead of following the previous path of time-consuming numerical and empirical formulations based on field experiments, the authors decided to continue the spectrum from some specific frequency point, well above the spectral peak, with the Phillips law $\sim \omega^{-5}$, which decays

faster than equilibrium spectrum $\omega^{-4}$, and therefore corresponds to a net wave energy absorption. Although a version of this concept was incorporated by Janssen (2009), detailed forms of this source term have not been developed to date, other than that spectrum at high frequencies appears to consistently tend toward an $\sim \omega^{-5}$ form as noted by Phillips (1985). Additional evidence for a transition from $\sim \omega^{-4}$ to $\sim \omega^{-5}$ at frequencies above the equilibrium range comes from analysis of multiple data sets by Resio et al. (2004a). In that paper the transition from $\sim \omega^{-4}$ to $\sim \omega^{-5}$ occurs approximately at $f_d = 1.1$ Hz, i.e.

the physical spectrum has to be continued from this point by $\sim \omega^{-5}$.

The spectrum amplitude at the junction frequency $f_d$ is dynamically changing in time. It is important that this analytic continuation contributes to a differential in inverse action, which also affects frequencies lower than $f_d$, since the nonlinear interaction term $S_{nl}$ is calculated over both "dynamic" and fixed Phillips areas. Therefore, the Phillips part of the spectrum "sends" the information about presence of the dissipation above $f_d$ to the rest of the spectrum.

At this point, all that remains for source-term closure in the HE model, is the coefficient in front of the wind input term, since it is not well defined experimentally. If we carry the numerical simulation with some arbitrary chosen coefficient, we could obtain a range of spectral energies, but would retain the qualitative properties of HE, like $\sim \omega^{-4}$ spectrum, spectral peak down-shift and peak frequency behavior in accordance with self-similar laws, but will miss the level of the total wave spectral energy.

To solve this, we choose the wind source coefficient to reproduce the same wave energy growth as was observed in field experiments. The value of this coefficient, found from the comparison with field observations wave energy growth is equal 0.05. This step completes the construction of the HE model.

In the next sections we proceed with numerical simulations based on described above HE model. Section 5 discusses the details of numerical model setup. Section 6 describes duration limited numerical simulation, which is the subject of more

academic than applied interest, targeted at self-similarity concept support, while the limited fetch numerical simulation results, described in section 7, besides academic interest, are the subject of comparison with the field experiments. A check of the compliance of numerical results with field experimental measurements is presented in the section 8.

## 2   Experimental evidence

Here we examine the empirical evidence from around the world, which has been utilized to quantify energy levels within the

equilibrium spectral range by Resio et al. (2004a). For convenience, we shall also use the same notation used by Resio et al. (2004a) in their study, for the angular averaged spectral energy densities in frequency and wavenumber spaces:

$$E_4(f) = \frac{2\pi \alpha_4 V g}{(2\pi f)^4} \tag{6}$$
$$F_4(k) = \beta k^{-5/2} \tag{7}$$

where $f = \frac{\omega}{2\pi}$, $\alpha_4$ is the constant, $V$ is some characteristic velocity and $\beta = \frac{1}{2}\alpha_4 V g^{-1/2}$. These notations are based on relation

of spectral densities $E(f)$ and $F(k)$ in frequency $f = \frac{\omega}{2\pi}$ and wave-number $k$ bases:

$$F(k) = \frac{c_g}{2\pi} E(f) \tag{8}$$

where $c_g = \frac{d\omega}{dk} = \frac{1}{2 \cdot 2\pi} \frac{g}{f}$ is the group velocity.

The notations in Eqs.(6)-(7) are connected with the spectral energy density $\epsilon(\omega, \theta)$ through

$$E(f) = 2\pi \int_0^{2\pi} \epsilon(\omega, \theta) d\theta \tag{9}$$

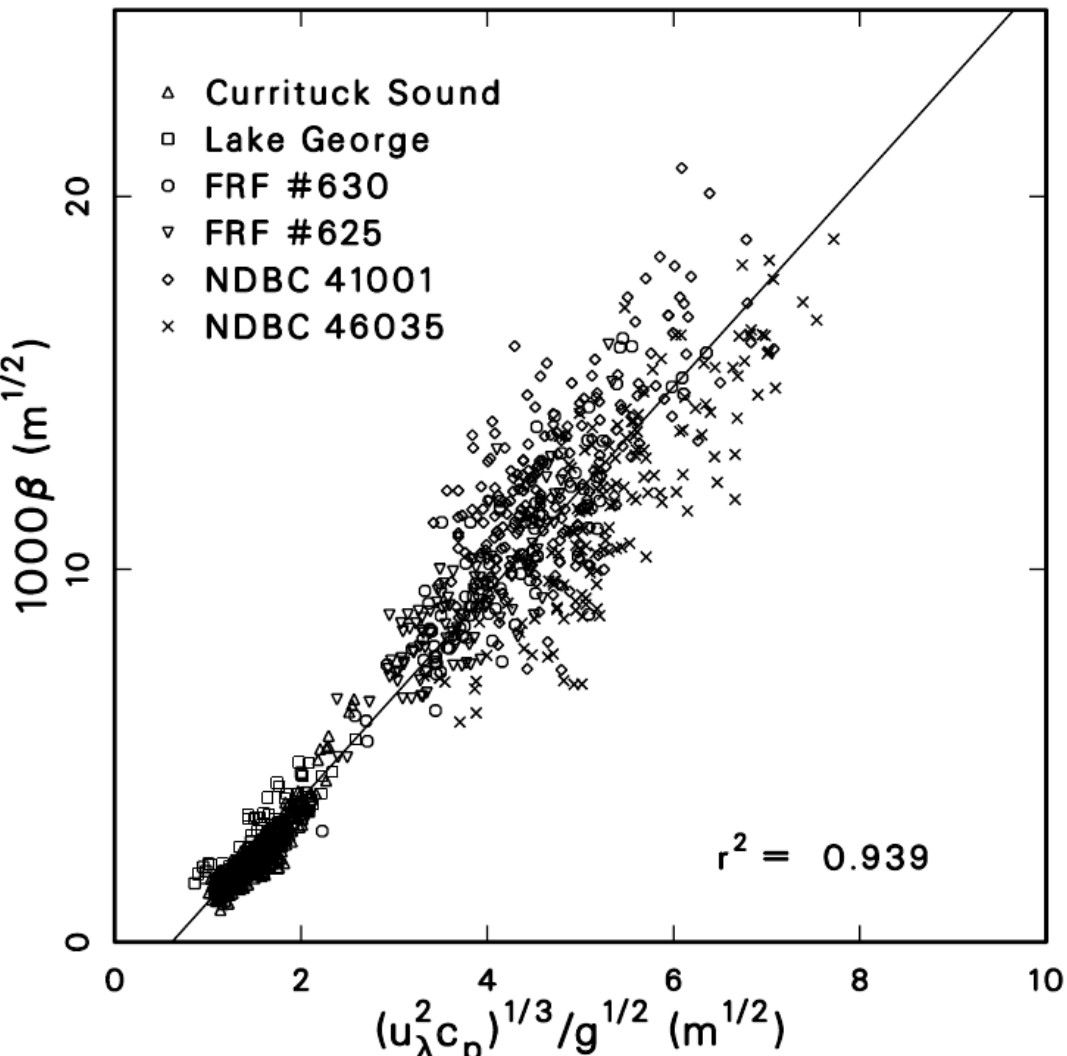

**Figure 1.** Correlation of equilibrium range coefficient $\beta$ with $(u_\lambda^2 c_p)^{1/3}/g^{1/2}$ based on data from six disparate sources. Adapted from Resio et al. (2004a)

The Resio et al. (2004a) analysis showed that experimental energy spectra $F(k)$, estimated through averaging $< k^{5/2} F(k) >$, can be approximated by linear regression line as the function of $(u_\lambda^2 c_p)^{1/3} g^{-1/2}$. Fig.1 shows that the regression line

$$\beta = \frac{1}{2}\alpha_4 \left[ (u_\lambda^2 c_p)^{1/3} - u_0 \right] g^{-1/2} \tag{10}$$

indeed, seems to be a reasonable approximation of these observations.

5      Here $\alpha_4 = 0.00553$, $u_0 = 1.93\ m/sec$, $c_p$ is the spectral peak phase speed and $u_\lambda$ is the wind speed at the elevation equal to a fixed fraction $\lambda = 0.065$ of the spectral peak wavelength $2\pi/k_p$, where $k_p$ is the spectral peak wave number. It is important

to emphasize that Resio et al. (2004a) experiments show that parameter $\beta$ increases with development of the wind-driven sea, when $f_p$ decreases and $C_p$ increases. This observation is consistent with the weak turbulent theory, where $\beta \sim P^{1/3}$ (Zakharov et al., 1992); here $P$ is the wave energy flux toward small scales.

Resio et al. (2004a) assumed that the near surface boundary layer can be treated as neutral and thus follows a conventional logarithmic profile

$$u_\lambda = \frac{u_\star}{\kappa} \ln \frac{z}{z_0} \tag{11}$$

having Von Karman coefficient $\kappa = 0.41$, where $z = \lambda \cdot 2\pi/k_p$ is the elevation equal to a fixed fraction $\lambda = 0.065$ of the spectral peak wavelength $2\pi/k_p$, where $k_p$ is the spectral peak wave number, and $z_0 = \alpha_C u_\star^2/g$ subject to Charnock (1955) surface roughness with $\alpha_C = 0.015$.

## 3 Theoretical considerations

Self-similar solutions consistent with the conservative kinetic equation

$$\frac{\partial \epsilon(\omega, \theta)}{\partial t} = S_{nl} \tag{12}$$

were studied in Zakharov (2005), Badulin et al. (2005). In this section we study self-similar solutions of the forced kinetic equation

$$\frac{\partial \epsilon(\omega, \theta)}{\partial t} = S_{nl} + \gamma(\omega, \theta)\epsilon(\omega, \theta) \tag{13}$$

where $\epsilon(\omega, \theta) = \frac{2\omega^4}{g} N(\mathbf{k}, \theta)$ is the energy spectrum.

One should note that this equation does not contain any explicit wave dissipation term, the role of dissipation is played by an existence of the energy sink at high wave numbers, in spirit of the WTT, see Zakharov and Filonenko (1967), Zakharov et al. (1992).

For our purposes, it is sufficient to simply use the dimensional estimate for $S_{nl}$,

$$S_{nl} \simeq \omega \left( \frac{\omega^5 \epsilon}{g^2} \right)^2 \epsilon \tag{14}$$

Eq.(13) has a self-similar solution if

$$\gamma(\omega, \theta) = \alpha \omega^{1+s} f(\theta) \tag{15}$$

where $s$ is a constant. Looking for self-similar solution in the form

$$\epsilon(\omega, t) = t^{p+q} F(\omega t^q) \tag{16}$$

we find

$$q = \frac{1}{s+1} \tag{17}$$

$$p = \frac{9q - 1}{2} = \frac{8 - s}{2(s+1)} \tag{18}$$

The function $F(\xi)$ has the maximum at $\xi \sim \xi_p$, thus the frequency of the spectral peak is

$$\omega_p \simeq \xi_p t^{-q} \tag{19}$$

The phase velocity at the spectral peak is

$$c_p = \frac{g}{\omega_p} = \frac{g}{\xi_p} t^q = \frac{g}{\xi_p} t^{\frac{1}{s+1}} \tag{20}$$

According to experimental data, the main energy input into the spectrum occurs in the vicinity of the spectral peak, i.e. at $\omega \simeq \omega_p$. For $\omega >> \omega_p$, the spectrum is described by Zakharov-Filonenko tail

$$\epsilon(\omega) \sim P^{1/3}\omega^{-4} \tag{21}$$

Here

$$P = \int\limits_{0}^{\infty}\int\limits_{0}^{2\pi} \gamma(\omega,\theta)\epsilon(\omega,\theta)d\omega d\theta \tag{22}$$

This integral converges if $s < 2$. For large $\omega$

$$\epsilon(\omega,t) \simeq \frac{t^{p-3q}}{\omega^4} \simeq \frac{t^{\frac{2-s}{2(s+1)}}}{\omega^4} \tag{23}$$

More accurately

$$\epsilon(\omega,t) \;\simeq\; \frac{\mu g}{\omega^4}u^{1-\eta}c_p^{\eta}g(\theta) \tag{24}$$

$$\eta \;=\; \frac{2-s}{2} \tag{25}$$

Now supposing $s = 4/3$ and $\gamma \simeq \omega^{7/3}$, we get $\eta = 1/3$ , which is exactly experimental regression line prediction. Because it is known from regression line in Fig.1 that $\xi = 1/3$, we immediately get $s = 4/3$ and the wind input term

$$S_{wind} \simeq \omega^{7/3}\epsilon \tag{26}$$

For many years, the assumption has been that there could be a net input or dissipation within the equilibrium range; however, Thomson et al. (2013) recently used extensive data from Ocean Station Papa to show that there was minimal wind input into
the wave spectrum in the equilibrium range. Resio et al. (2004a) suggest that the existence of significant net energy input or dissipation within the frequency range would tend to force the spectrum away from an $f^{-4}$ form, contrary to the pattern found in field measurements. If we assume that the wind source is primarily centered on the spectral peak, the only missing component in our numerical solution is an unknown coefficient in front of it, which will be defined later from the comparison with total energy growth in experimental observations.

Another important theoretical relationship, that can be derived from joint consideration of Eqs. (6), (8) and (24) is

$$1000\beta = \lambda\frac{(u^2 c_p)^{1/3}}{g^{1/2}} \tag{27}$$

which shows a theoretical equivalence to the experimental regression, where $\lambda$ is an unknown constant, defined experimentally.

At the end of the section, we present the summary of important relationships.

Wave action $N$, energy $E$ and momentum $M$ in frequency-angle presentation are:

$$N = \frac{2}{g^2} \int_0^\infty \int_0^{2\pi} \omega^3 n \, d\omega \, d\phi \tag{28}$$

$$E = \frac{2}{g^2} \int_0^\infty \int_0^{2\pi} \omega^4 n \, d\omega \, d\phi \tag{29}$$

$$M = \frac{2}{g^3} \int_0^\infty \int_0^{2\pi} \omega^5 n \cos\phi \, d\omega \, d\phi \tag{30}$$

The self-similar relations for duration limited case are given by:

$$\epsilon = t^{p+q} F(\omega t^q) \tag{31}$$

$$9q - 2p = 1, \ p = 10/7, \ q = 3/7 \ s = 4/3 \tag{32}$$

$$N \sim t^{p+q} \tag{33}$$

$$E \sim t^p \tag{34}$$

$$M \sim t^{p-q} \tag{35}$$

$$<\omega> \sim t^{-q} \tag{36}$$

The same sort of self-similar analysis gives self-similar relations for fetch limited case:

$$\epsilon = \chi^{p+q} F(\omega \chi^q) \tag{37}$$

$$10q - 2p = 1, \ p = 1, \ q = 3/10 \ s = 4/3 \tag{38}$$

$$N \sim \chi^{p+q} \tag{39}$$

$$E \sim \chi^p \tag{40}$$

$$M \sim \chi^{p-q} \tag{41}$$

$$<\omega> \sim \chi^{-q} \tag{42}$$

## 4   The details of "implicit" dissipation

Now that the construction of ZRP wind input term with the unknown coefficient has been accomplished in spirit of WTT in the previous chapter, the HE model, suitable for numerical simulation still misses dissipation term, localized at finite wave numbers

– there is no such thing as the infinite phase volume in the reality: the real ocean Fourier space is confined by characteristic wave number, corresponding to the start of the dissipation effects, caused by the wave-breaking events.

There is a lot of freedom in choosing the dissipation term. Since there is no current interpretation of the wave-breaking dissipation mechanism, one can choose it in whatever shape it is preferred, but any particular choice will be questioned since it is an artificial one.

Because of that, the motivation consisted in the fact that at the current "proof if the concept" stage one need to know the effective sink with the simplest structure. Continuation of the spectrum from $\omega_d$ with Phillips law $A(\omega_d) \cdot \omega^{-5}$ (see Phillips (1966)), decaying faster than equilibrium spectrum $\omega^{-4}$, will get high-frequency dissipation. The corresponding analytic parameterization of this dissipation term will be unknown, while not in principle impossible to figure out in some way. One should note that this method of dissipation is not our invention, it is described in Janssen (2009).

Specifically, the coefficient $A(\omega_d)$ in front of $\omega^{-5}$, is unknown, but is not required to be defined in an explicit form. Instead, it is dynamically determined from the continuity condition of the spectrum, at frequency $\omega_d$, on every time step. In other words, the starting point of the Phillips spectrum coincides with the last point of the dynamically changing spectrum, at the frequency point $\omega_d = 2\pi f_d$, where $f_d \simeq 1.1\,Hz$, as per Long and Resio (2007). This is the way the high frequency "implicit" damping is incorporated into the alternative computational framework of $HE$. The question of the finer details of the high-frequency "implicit" damping structure is of secondary importance, at the current "proof of the concept" stage.

The whole set of the input and dissipation terms is accomplished now with one uncertainty: the explained approach leaves one parameter arbitrary – the constant in front of the wind input term. We choose it equal to 0.05 from the condition of the reproduction of the field observations of wave energy growth along the fetches, analyzed in Badulin et al. (2007).

## 5   Numerical validation of relationship

To check the self-similar hypothesis posed in Eq.(26), we performed a series of numerical simulations of Eq.(1) in the spatially homogeneous duration limited $\frac{\partial N}{\partial r} = 0$ and spatially inhomogeneous fetch limited $\frac{\partial N}{\partial t} = 0$ situations.

All simulations used WRT (Webb-Resio-Tracy) method (see Tracy and Resio (1982)), which calculates the nonlinear interaction term in the exact form. The presented numerical simulation utilized the version of WRT method, previously used in Webb (1978); Resio and Perrie (1989); Perrie and Zakharov (1999); Pushkarev et al. (2003); Resio et al. (2004b); Long and Resio (2007); Korotkevich et al. (2008); Badulin and Zakharov (2012); Pushkarev and Zakharov (2016), and used the grid of 71 logarithmically spaced points in the frequency range from $0.1Hz$ to $2.0Hz$ and 36 equidistant points in the angle domain. The constant time step in the range between 1 and 2 sec has been used for explicit first order accuracy order integration in time.

There is the balance, between the number of nodes of the grid and the volume of calculation to be performed. The particular version of WRT model has been tuned to the minimum grid number of nodes to solve realistic physical problems, but still be fast enough to simulate them over a reasonable time span. The correctness of this set-up is confirmed by multiple cited above numerical experiments, reproducing mathematical properties of Hasselmann equation.

For the convenience, we present the pseudo-code used for main cycle of the described model:

1. Calculate $S_{nl}(\varepsilon(f,\theta))$

2. Overwrite $\varepsilon(f,\theta)$ to $f^{-5}$ for $f > 1.1$ Hz

3. Update $\varepsilon(f,\theta) = \varepsilon(f,\theta) + dt \cdot S_{nl}(f,\theta)$

4. Solve analytically $\frac{\partial(f,\theta)}{\partial t} = S_{wind}(f,\theta)\varepsilon(f,\theta)$ for time $dt$

5. Return to step 1

All numerical simulations discussed in the current paper have been started from uniform noise energy distribution in Fourier space $\varepsilon(\omega,\theta) = 10^{-6}$ $m^4$, corresponding to small initial wave height with effectively negligible nonlinearity level. The constant wind of speed 10 m/sec was assumed blowing away from the shore line, along the fetch. The assumption of the constant wind speed is a necessary simplification, due to the fact that the numerical simulation is being compared to various data from field experiments, and the considered set-up is the simplest physical situation, which can be modeled.

The same ZRP wind input term Eq.(26) has been used in both cases in the form

$$S_{in}(\omega,\theta) = \gamma(\omega,\theta) \cdot \varepsilon(\omega,\theta) \tag{43}$$

$$\gamma(\omega,\theta) = \begin{cases} 0.05\frac{\rho_{air}}{\rho_{water}}\omega\left(\frac{\omega}{\omega_0}\right)^{4/3} q(\theta) & \text{for } f_{min} \leq f \leq f_d, \ \omega = 2\pi f \\ 0 & \text{otherwise} \end{cases} \tag{44}$$

$$q(\theta) = \begin{cases} \cos 2\theta & \text{for } -\pi/4 \leq \theta \leq \pi/4 \\ 0 & \text{otherwise} \end{cases} \tag{45}$$

$$\omega_0 = \frac{g}{U}, \ \frac{\rho_{air}}{\rho_{water}} = 1.3 \cdot 10^{-3} \tag{46}$$

where $U$ is the wind speed at the reference level of 10 meters, $\rho_{air}$ and $\rho_{water}$ are the air and water density correspondingly. It is conceivable to use more sophisticated expression for $q(\theta)$, for instance $q(\theta) = q(\theta) - q(0)$. To make direct comparison with experimental results of Resio et al. (2004a), we used the relation $u_\star \simeq U/28$ (see Golitsyn (2010)) in Eq.(11). Frequencies $f_{min}$ and $f_d$ depend on the wind speed and should be found empirically. In current numerical experiments for $U = 10$ m/sec and $U = 5$ m/sec, $f_{min} = 0.1$ Hz and $f_d = 1.1$ Hz. This choice is justified by obtained numerical results.

The above described "implicit dissipation" term $S_{diss}$ has played dual role of direct energy cascade flux sink due to wave breaking as well as numerical scheme stabilization factor at high wave-numbers.

## 6   Duration limited numerical simulation

The duration limited simulation has been performed for a wind speed of $U = 10$ m/sec.

Fig.2 shows the total energy growth as the function of time, consistent with self-similar prediction Eq.(34) for index $p = 10/7$, supplied with the empirical coefficient in front of it, see Fig.3.

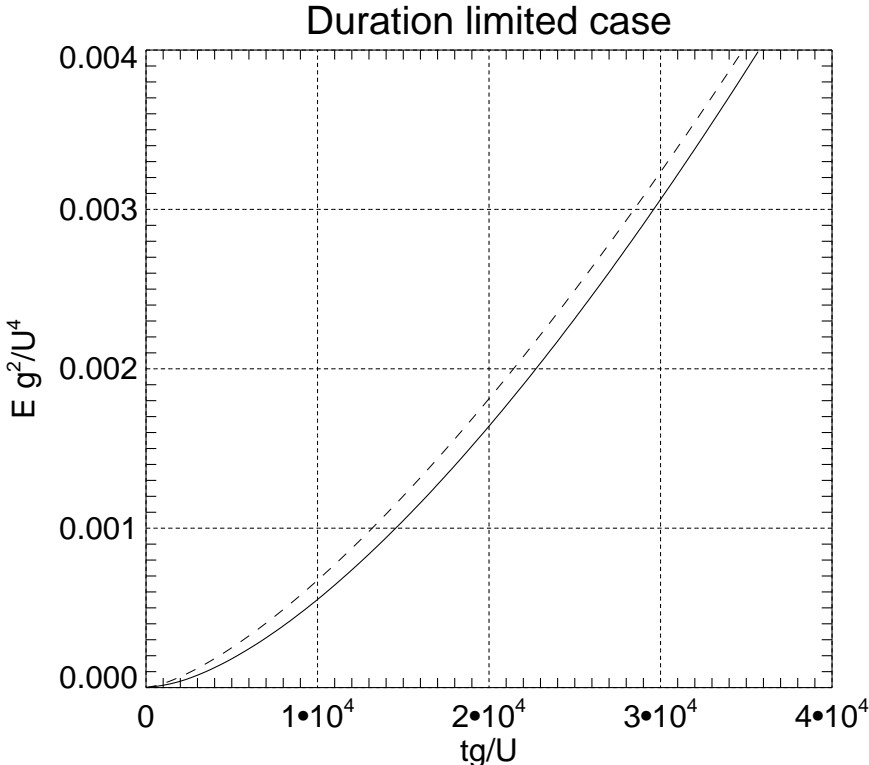

**Figure 2.** Dimensionless energy $Eg^2/U^4$ versus dimensionless time $tg/U$ for wind speed $U = 10$ m/sec duration limited case - solid line. Self-similar solution with the empirical coefficient in front of it: $1.3 \cdot 10^{-9} \left(tg/U\right)^{10/7}$ - dashed line.

One should specifically elaborate on the local index $p$ numerical calculation procedure for Fig.3. First, the total energy function was smoothed via moving average, then the corresponding derivative is estimated numerically via finite differences, and finally a moving average is used to obtain the time-varying index value.

The relatively small systematic deviation from self-similar behavior, visible on Fig.2, is connected with the following two

5 facts.

First, the transition process in the beginning of the simulation, when the wave system behavior is far from self-similar one. The self-similar solution is pure power function, which doesn't take into account the initial transition process, and that causes the systematic difference. This systematic difference could be diminished via a parallel shift, which would take into account the initial transition process. Such parallel shift is equivalent to starting the simulation from different initial conditions.

10 The second fact is the asymptotic nature of the self-similar solution, producing an evolution of the simulated wave system toward self-similar behavior with increasing time. As seen on Fig.3, the numerical value of the local exponent converges to the theoretical value $p = 10/7$, reaching approximately $6\%$ accuracy for sufficient dimensionless time $3 \cdot 10^4$ .

The dependence of the mean frequency on time, shown on Fig.4, is consistent with the self-similar dependence found in Eq.(36) for $q = 3/7$, supplied with the empirical coefficient in front of it, see Fig.5.

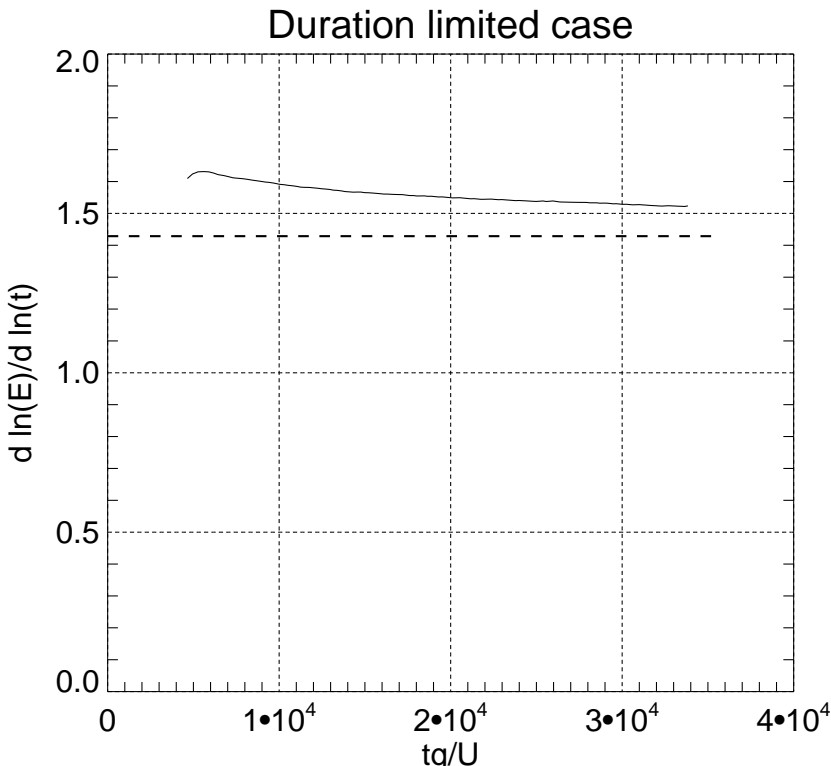

**Figure 3.** Energy local power function index $p = \frac{d \ln E}{d \ln t}$ as a function of dimensionless time $tg/U$ for wind speed $U = 10$ m/sec duration limited case - solid line. Theoretical value of self-similar index $p = 10/7$ - thick horizontal dashed line.

The systematic deviation of two lines on Fig.4, remain within $3\%$ of the target value $q = 3/7$ for the same reasons as for wave energy behavior - the transition process in the beginning of the simulation and asymptotic nature of self-similar solution.

A check of the consistency with the "magic number" $9q - 2p = 1$ (see Eq.(32) ), is presented on Fig.6. The reason of systematic deviation from the target value 1 is obviously connected with the reasons of the systematic deviations of $p$ and $q$, as the "magic number" is calculated as their linear combination, reaching the accuracy of approximately $10\%$ for a long enough dimensionless time of $3 \cdot 10^4$.

One should note that indices $p, q$ and the "magic relation" $9q - 2p$ exhibit asymptotic convergence to the corresponding target values.

Fig.7 presents angle-integrated energy spectrum, as the function of frequency, in logarithmic coordinates. One can see that it consists of the segments of:

- the spectral peak region

- the inertial (equilibrium) range $\omega^{-4}$ spanning from the spectral peak to the beginning of the "implicit dissipation" $f_d = 1.1$ Hz

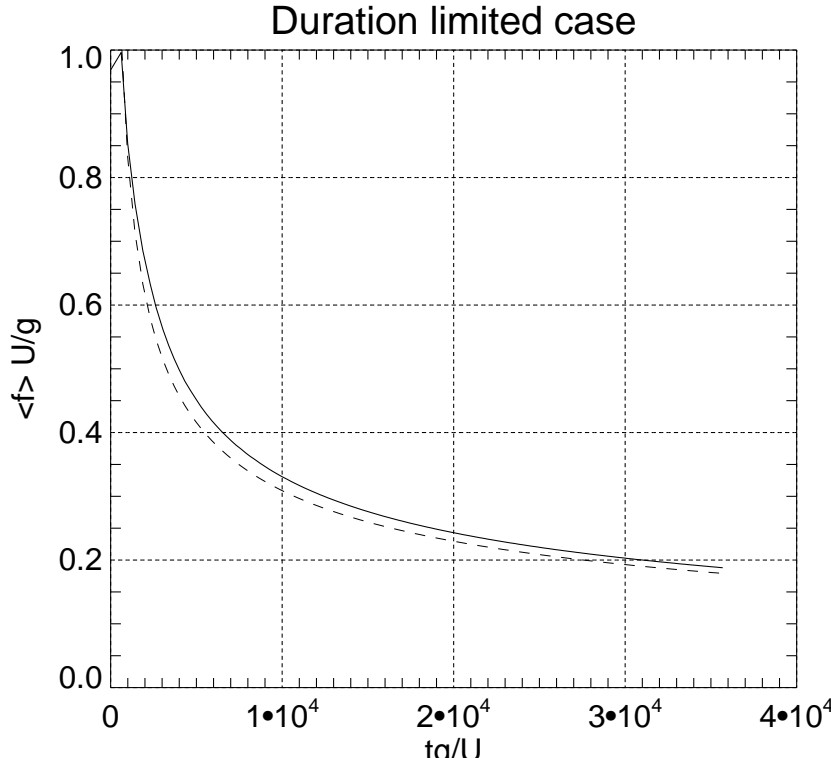

**Figure 4.** Dimensionless mean frequency $< f > \cdot U/g = E/N \cdot U/g$ (solid line) versus dimensionless time $tg/U$ for wind speed $U = 10$ m/sec duration limited case - solid line, self-similar solution with the empirical coefficient in front of it: $16.0 \cdot (tg/U)^{-3/7}$ - dashed line.

– Phillips high frequency tail $\omega^{-5}$ starting approximately from $f_d = 1.1$ Hz

The compensated spectrum $F(k) \cdot k^{5/2}$ is presented in Fig.8.

One can see plateau-like region responsible for $k^{-5/2}$ behavior, equivalent to $\sim f^{-4}$ tail in Fig.7. This shape of the spectrum is similar to observed by Resio et al. (2004b); Long and Resio (2007). This exact solution of Eq.(12), known as KZ spectrum, was found by Zakharov and Filonenko (1967). The universality of $f^{-4}$ asymptotic for the "inertial" (also known as "equilibrium" in oceanography) range between spectral peak energy input and high-frequency energy dissipation areas has been observed in multiple experimental field observations and is accepted by the oceanographic community after the seminal work of Phillips (1985). One should note that most of the energy flux into the system comes in the vicinity of the spectral peak, as shown in Fig.9, providing significant inertial interval for KZ spectrum.

The angular spectral distribution of energy, presented in Fig.10, is consistent with the results of experimental observations Resio et al. (2011), that show a broadening of the angular spreading in both directions away from the spectral peak frequency.

To compare the duration limited numerical simulation results with the experimental analysis by Resio et al. (2004a), presented in Fig.1, Fig.11 shows the function $\beta = F(k) \cdot k^{5/2}$ as the function of $(u_\lambda^2 C_p)^{1/3}/g^{1/2}$ for wind speed $U = 10$ m/sec, along with the regression line from Resio et al. (2004a) and theoretical prediction Eq.(27) for $\lambda = 2.74$ . The numerical results

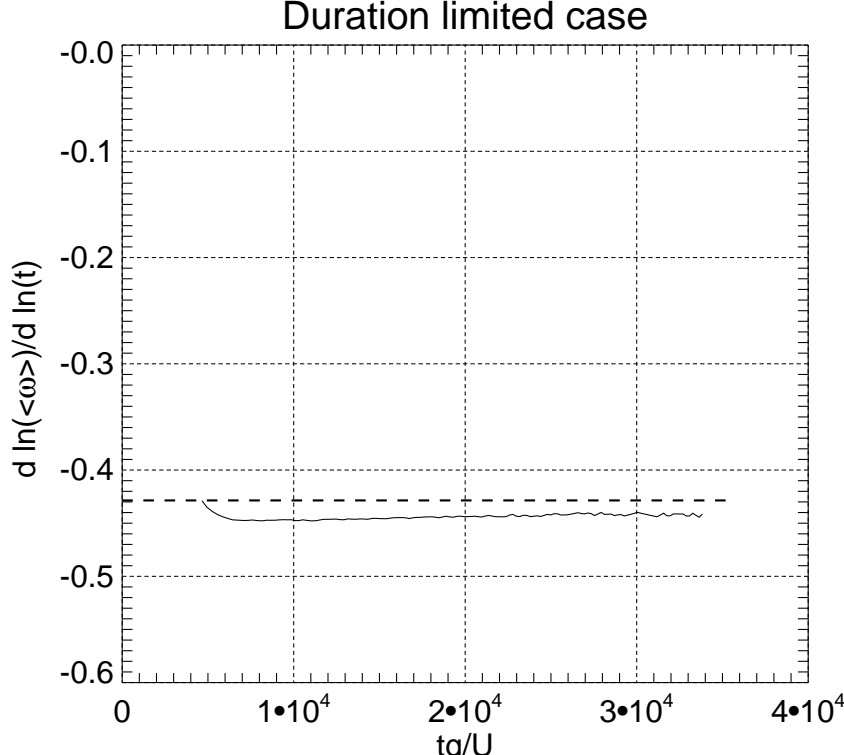

**Figure 5.** Mean frequency local power function index $-q = \frac{d\ln<\omega>}{d\ln t}$ as the function of dimensionless time $tg/U$ for wind speed $U = 10$ m/sec duration limited case (solid line). Theoretical value of self-similar exponent $q = -3/7$ - thick horizontal dashed line.

and theoretical prediction line fall within a very small RMS deviation ($r^2 = 0.939$, see Fig.1) from the regression line. One should note asymptotic convergence of the numerical simulation results to the theoretical line.

## 7 Limited fetch numerical simulation

The limited fetch simulation was performed in the framework of the stationary version of Eq.(1):

$$5 \quad \frac{1}{2}\frac{g\cos\theta}{\omega}\frac{\partial\epsilon}{\partial x} = S_{nl}(\epsilon) + S_{wind} + S_{diss} \tag{47}$$

where $x$ is chosen as the coordinate axis orthogonal to the shore and $\theta$ is the angle between individual wavenumber $\boldsymbol{k}$ and the axis $\boldsymbol{x}$. To find the dependence on the wind speed, directed off the shore, two numerical simulations for wind speeds of $U = 5\mathrm{m/sec}$ and $U = 10\mathrm{m/sec}$ have been performed.

The stationarity in Eq.(47) is somewhat difficult for numerical simulation, since it contains a singularity in the form of $\cos\theta$

10   in front of $\frac{\partial\epsilon}{\partial x}$. This problem was overcome by zeroing one half of the Fourier space of the system for the waves propagating

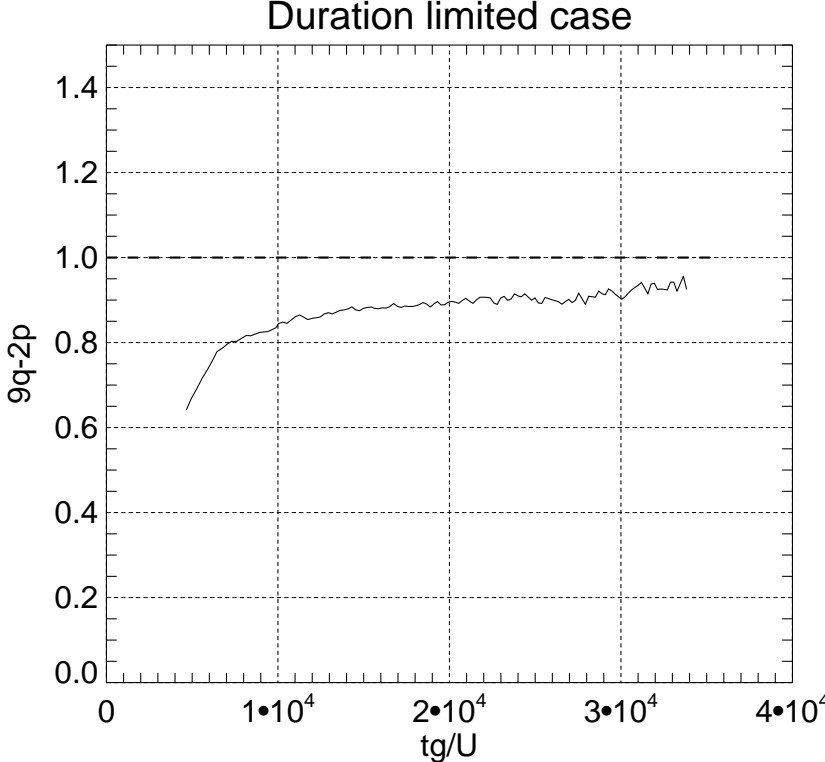

**Figure 6.** "Magic number" $9q - 2p$ as the function of dimensionless time $tg/U$ for wind speed $U = 10$ m/sec duration limited case - solid line. The target value $1$ for the self-similar relation Eq.(32) is presented by the horizontal dashed line.

toward the shore. Since the energy in such waves is small with respect to waves propagating in the offshore direction, such an approximation is quite reasonable for our purposes.

Since the wind forcing index $s$ in the fetch-limited case is similar to that in the duration limited case, the numerical simulation of Eq.(47) has been performed for the same input functions as in the duration limited case with the same low-level energy noise initial conditions in Fourier space.

Fig.12 presents total energy growth as a function of fetch, consistent with self-similar solution Eq.(40) for index $p = 1$, and its appropriate empirical coefficient. The corresponding values of indices $p$ along the fetch are presented on Fig.13. The small amplitude oscillations observed in the index behavior can be attributed to the finite grid resolution used in the simulation.

As it was noted above, the wave evolution for wind speed $U = 5$ m/sec case is expected to be slower than for $U = 10$ m/sec case due to weaker nonlinear interaction term. One can see, indeed, slower asymptotic convergence of the calculated total energy local power index to the target value $p = 1$ for $U = 5$ m/sec case, being compared to $U = 10$ m/sec case. The deviation of results from the $U = 10$ m/sec case relative to the target value does not exceed an error of about $5\%$, while for $U = 5$ m/sec case the error doesn't exceed $20\%$. The role of relatively short in time non-self-similar development of the wave system at the

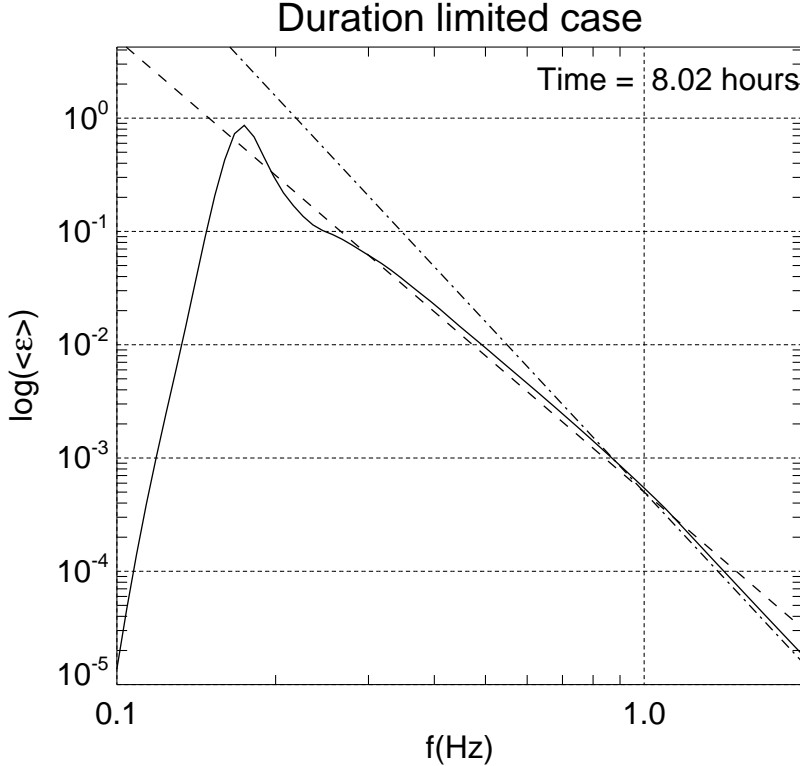

**Figure 7.** Decimal logarithm of the angle averaged spectrum as the function of the decimal logarithm of the frequency for wind speed $U = 10$ m/sec duration limited case - solid line. Spectrum $\sim f^{-4}$ - dashed line, spectrum $\sim f^{-5}$ - dash-dotted line.

very beginning of the fetch should be noted as well as the factor contributing to the deviation from the target value of index p=1: the wave system obviously needs some time to evolve into fully self-similar mode.

The dependence of the mean frequency fetch, shown on Fig.14, is consistent with the self-similar dependence Eq.(42) for index $q = 0.3$, supplied with the empirical coefficient in front of it. The small amplitude oscillations observed in index behavior can be attributed to the finite grid resolution used in the simulation, since the spectral peak moves continuously between discrete frequencies in a manner that cannot be matched in these discretized simulations.

The local values of indices $q$ for two different wind speed amplitudes are presented on Fig.15 along with the target value of self-similar index $q = 0.3$. After sufficient fetch one can see only about $14\%$ deviation from the target value for $U = 10$ m/sec case and about $2.5\%$ for $U = 5$ m/sec case.

The reasons for the $10\%$ systematic deviation from the self-similar solutions observed in the lines on Fig.14, corresponding to the wind speeds of $U = 5$ m/sec and $U = 10$ m/sec, are the same as noted previously for wave energy behavior - the transition process in the beginning of the simulation and asymptotic nature of self-similar solution.

The check on the consistency of calculated "magic number" $(10q - 2p)$ (see Eq.(38)) is presented in Fig.16. The reason of systematic deviation from the target value 1 is obviously connected with the systematic deviations of $p$ and $q$, as the "magic

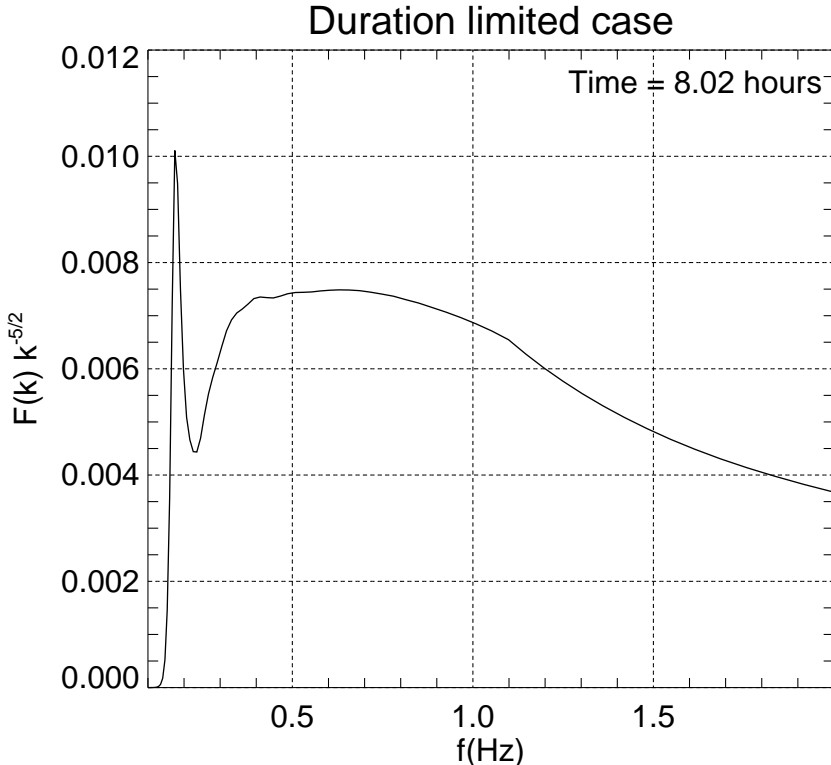

**Figure 8.** Compensated spectrum as the function of linear frequency $f$ for wind speed $U = 10$ m/sec duration limited case.

number" is calculated as their linear combination, reaching the accuracy of approximately $10\%$ for fetches $3 \cdot 10^4$. As noted previously, the small amplitude oscillations observed in the indices behavior can be attributed to the finite grid resolution used in the simulation.

One should note that indices $p, q$ and the "magic relation" $10q - 2p$ exhibit asymptotic convergence to the corresponding target values.

Fig.17 presents directionally-integrated energy spectrum, as the function of frequency, in logarithmic coordinates. As could be seen in the duration-limited case, one can see that it consists of three process-related segments:

- the spectral peak region

- the inertial (equilibrium) range $\omega^{-4}$ spanning from the spectral peak to the beginning of the "implicit dissipation" $f_d = 1.1$ Hz

- Phillips high frequency tail $\omega^{-5}$ starting approximately at $f_d = 1.1$ Hz

The compensated spectrum $F(k) \cdot k^{5/2}$ is presented in Fig.20. One can see plateau-like region responsible for $k^{-5/2}$ behavior, equivalent to $\omega^{-4}$ tail in Fig.17 and similar to observed by Resio et al. (2004b); Long and Resio (2007). As in the duration

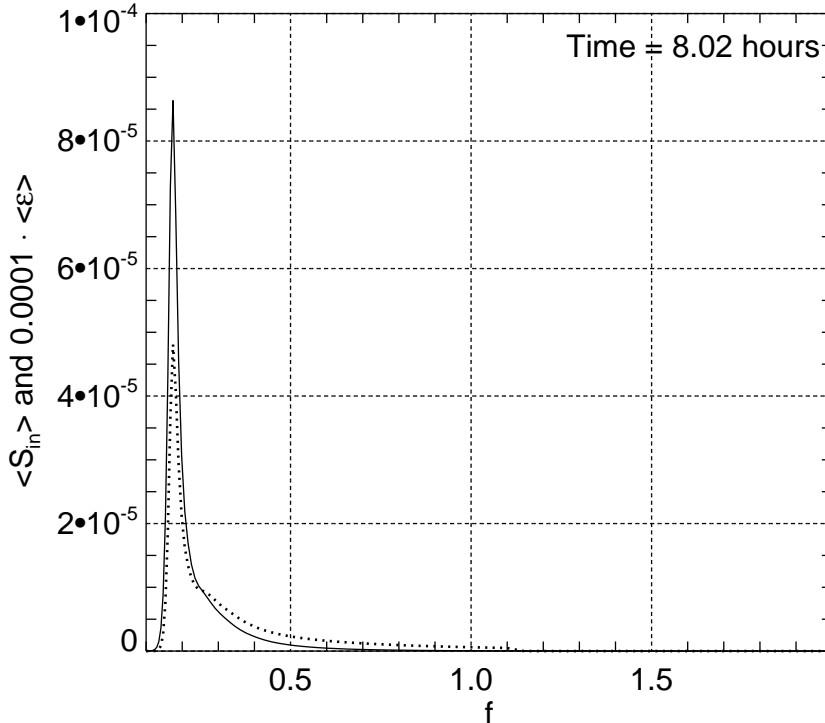

**Figure 9.** Typical, angle averaged, wind input function density $< S_{in} >= \frac{1}{2\pi} \int \gamma(\omega,\theta)\varepsilon(\omega,\theta)d\theta$ (dotted line) and angle averaged spectrum $< \varepsilon >= \frac{1}{2\pi} \int \varepsilon(\omega,\theta)d\theta$ (solid line) as the functions of the frequency $f = \frac{\omega}{2\pi}$ for wind speed $U = 10$ m/sec duration limited case.

limited case, the KZ solution (Zakharov and Filonenko, 1967) also holds for the fetch limited case, and most of the energy flux into the system comes in the vicinity of the spectral peak as well, as shown in Fig.18, providing significant inertial (equilibrium) range for KZ spectrum between spectral peak energy input and high-frequency energy dissipation areas .

The angular spectral distribution of energy, presented in Fig.21, as in the duration limited case, is consistent with the results of experimental observations by Resio et al. (2011), that show a broadening of the angular spreading in both directions away from the spectral peak frequency.

The excess spectral energy at very oblique angles is a numerical artifact, connected with the specifics of how the limited fetch statement is simulated in here, i.e. above mentioned singularity presence at the left hand side of the Eq.(47) at $\theta = \pm\pi/2$.

The detailed structure of angular spreading for both duration and limited fetch cases is given on Fig.19 . The time that would be required to produce such a pattern is far in excess of the time for this excess energy to be removed from the equilibrium range by the nonlinear flux and can be shown to vanish when a time-space simulation is used instead of the stationary solution assumed here.

It is clearly seen that the "blobs" in the limited fetch case contain no more than $5\%$ of the total energy of the corresponding spectrum and could be neglected for the purposes of the presented research.

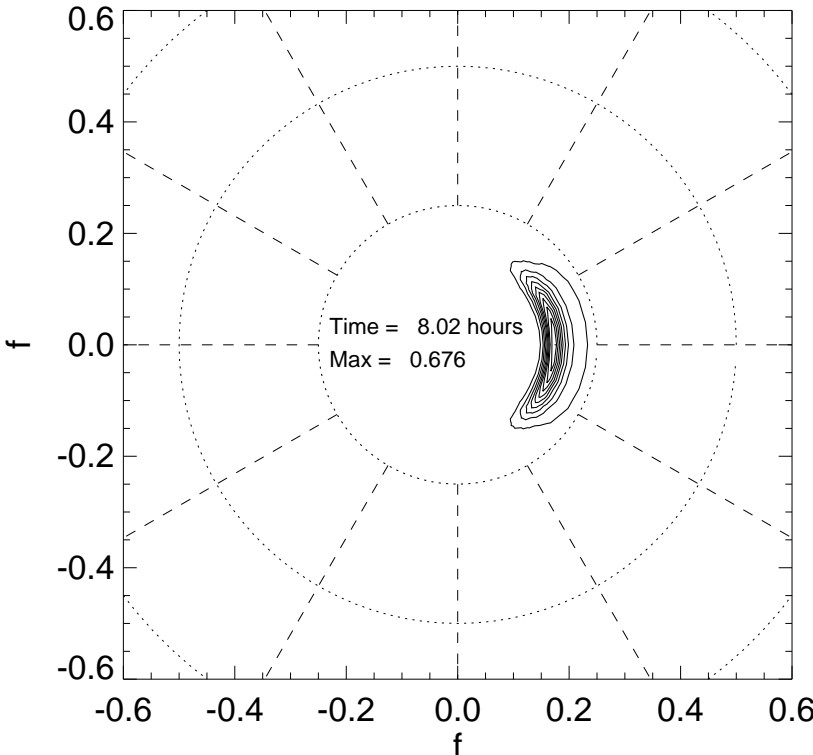

**Figure 10.** Angular spectra dependence for wind speed $U = 10$ m/sec duration limited case.

To compare the limited fetch numerical simulation results with the experimental analysis by Resio et al. (2004a), presented in Fig.1, Fig.22 shows the function $\beta = F(k) \cdot k^{5/2}$ as a function of $(u_\lambda^2 C_p)^{1/3}/g^{1/2}$ for wind speed $U = 10.0$ m/sec, along with the regression line from Resio et al. (2004a) and its theoretical prediction Eq.(27) for $\lambda = 2.11$. The numerical results and theoretical prediction line fall within RMS deviation ($r^2 = 0.939$, see Fig.1) from the regression line. One should note asymptotic convergence of the numerical simulation results to the theoretical line. Being parameterized by fetch coordinate, the numerical simulation results evolve from the left to the right on the graph, from the dimensionless fetch equal 0 to $3.0 \cdot 10^4$.

## 8   Comparison with the experiments

Comparison of limited fetch and duration limited simulations with the experimental results by Resio and Long (2007) and the theoretical prediction based on Eq.(10) is presented in Fig.11 and Fig.22. One should note that the numerical results and theoretical prediction line with corresponding values of $\lambda$ fall into the RMS deviation ($r^2 = 0.939$, see Fig.1) relative to the experimental regression line Eq.(10).

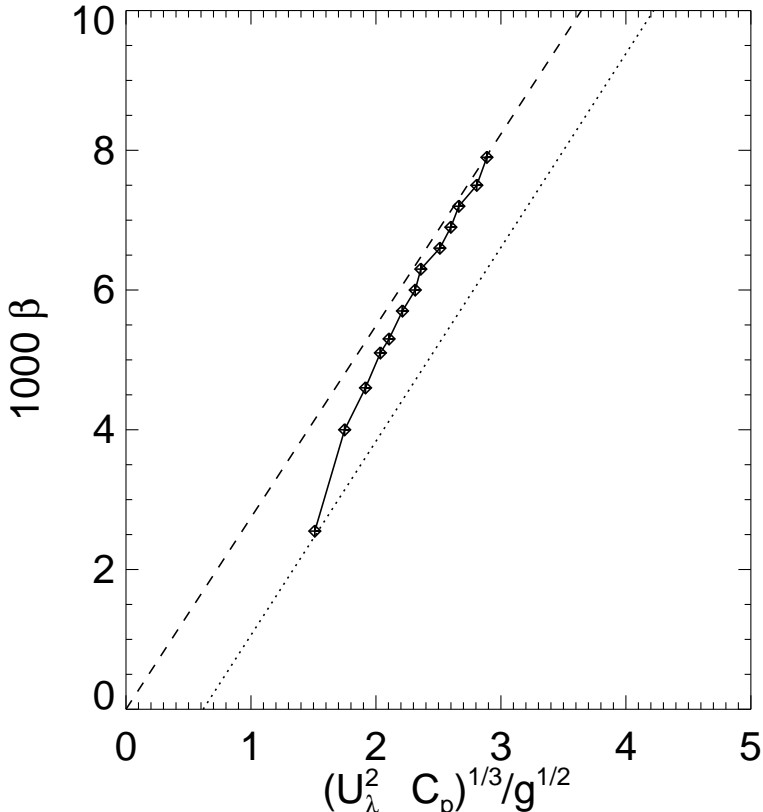

**Figure 11.** Experimental, theoretical and numerical evidence of the dependence of $1000\beta$ on $\left(u_\lambda^2 c_p\right)^{1/3}/g^{1/2}$. Dashed line - theoretical prediction Eq.(10) for $\lambda = 2.74$; dotted line - experimental regression line from Resio et al. (2004a); Resio and Long (2007). Line connected diamonds - results of numerical calculations for wind speed $U = 10$ m/sec duration limited case. Being parameterized by dimensionless time $tg/U$, the numerical simulation trajectory evolves from the left to the right on the graph, covering time span from $tg/U = 0$ to $tg/U \simeq 3.5 \cdot 10^5$.

The dependencies of the dimensionless energy and the frequency on the dimensionless fetch for the limited fetch simulation, superimposed on the experimental observations collected by Young (1999), are presented of Fig.23 and Fig.24, showing good consistency of the presented numerical results and the experimental observations.

## 9    Conclusions

5  We have analyzed the new ZRP form for wind input, proposed in Zakharov et al. (2012) in terms of both fetch-limited and duration-limited wave growth. The approach proposed here for the development of a set of balanced source terms uses only two empirical coefficients: one in the magnitude of the wind source term and the second in the location of the transition from $\sim \omega^{-4}$ to $\sim \omega^{-5}$ spectrum. This approach focuses on the combination of the theoretical finding of the self-similar solutions and the extraction of the relevant one through the comparison with the field experimental data.

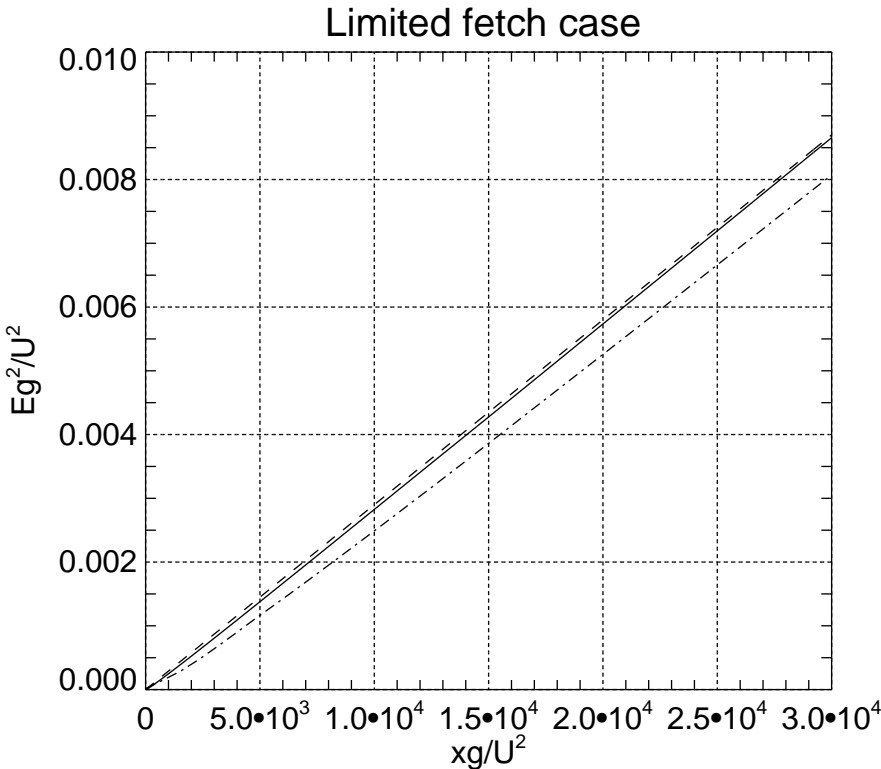

**Figure 12.** Dimensionless energy $Eg^2/U^4$ versus dimensionless fetch $xg/U^2$ for fetch limited case: wind speed $U = 10$ m/sec - solid line, wind speed $U = 5$ m/sec - dash-dotted line. Self-similar solution with the empirical coefficient in front of it: $2.9 \cdot 10^{-7} xg/U^2$ - dashed line.

 

The numerical simulations for both duration limited and fetch limited cases, using the ZRP wind input term, XNL nonlinear term $S_{nl}$ and "implicit" high-frequency dissipation, show remarkable consistency with predicted self-similar properties of HE and with the regression line from field studies, relating energy levels in the equilibrium range to wind speed by Resio et al. (2004a) and Resio and Long (2007).

5    The proposed model is the proof of the concept, providing strong support of simplified assumptions, such as discontinuity and the fixed frequency transition point of the source terms. The influence of these effects is planned to be studied , in particular, using more sophisticated approach by Zakharov and Badulin (2012) in the future.

Although the integral parameters of the model have been verified against the experimental observations, the verification of the spectral details, such as angular spreading, requires additional studies.

10   Observed oscillations of self-similar indices are interpreted as the effects of the discreteness of the model, which suggests that a study of the influence of the grid resolution on such oscillations is desirable in future research.

A test of the model invariance with respect to wind speed change from 5 to 10 m/sec has already been performed, but further study of the effects of wider range of wind speeds variation on self-similar properties of the model is desired in the future.

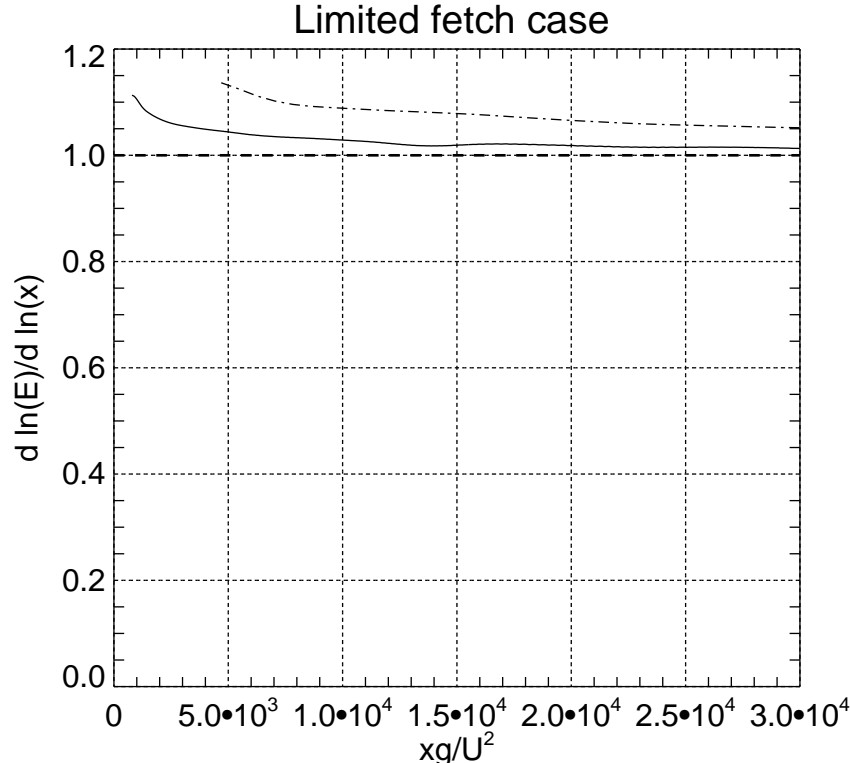

**Figure 13.** Energy local power function index $p = \frac{d\ln E}{d\ln x}$ as the function of dimensionless fetch $xg/U^2$ for fetch limited case: wind speed $U = 10$ m/sec - solid line, wind speed $U = 5$ m/sec - dash-dotted line. Theoretical value of self-similar index $p = 1$ - thick horizontal dashed line.

At the moment of submission of the manuscript, the main technical obstacle to effective development of new generation of physically based HE models is insufficiently fast calculation of the exact nonlinear interaction. The transition to 2D case requires radical increase of the calculations speed. We hope that such improvements will be made in near future.

The authors hope that this new framework will offer additional guidance for the source terms in operational models .

5   *Acknowledgements.*  The research presented in the section 6 has been accomplished due to the support of the grant "Wave turbulence: the theory, mathematical modeling and experiment" of the Russian Scientific Foundation No 14-22-00174. The research set forth in the section 1, was funded by the program of the presidium of RAS "Nonlinear dynamics in mathematical and physical sciences". The research presented in other chapters was supported by ONR grant N00014-10-1-0991.

The authors gratefully acknowledge the support of these foundations.

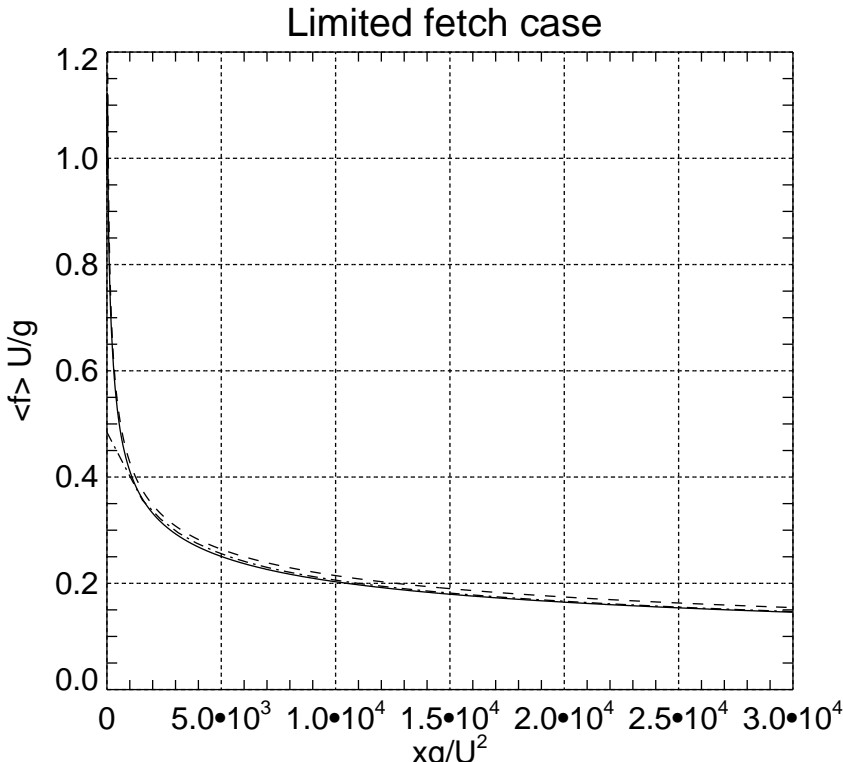

**Figure 14.** Dimensionless mean frequency, as the function of dimensionless fetch, calculated as $<f>= \frac{1}{2\pi} \frac{\int \omega n d\omega d\theta}{\int n d\omega d\theta}$, where $n(\omega,\theta) = \frac{\varepsilon(\omega,\theta)}{\omega}$ is the wave action spectrum, for wind speed 10 m/sec (solid line) and 5 m/sec (dashed line). The dash-dotted line is the self-similar dependence $3.4 \cdot \left(\frac{xg}{U^2}\right)^{-0.3}$ with the empirical coefficient in front of it.

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

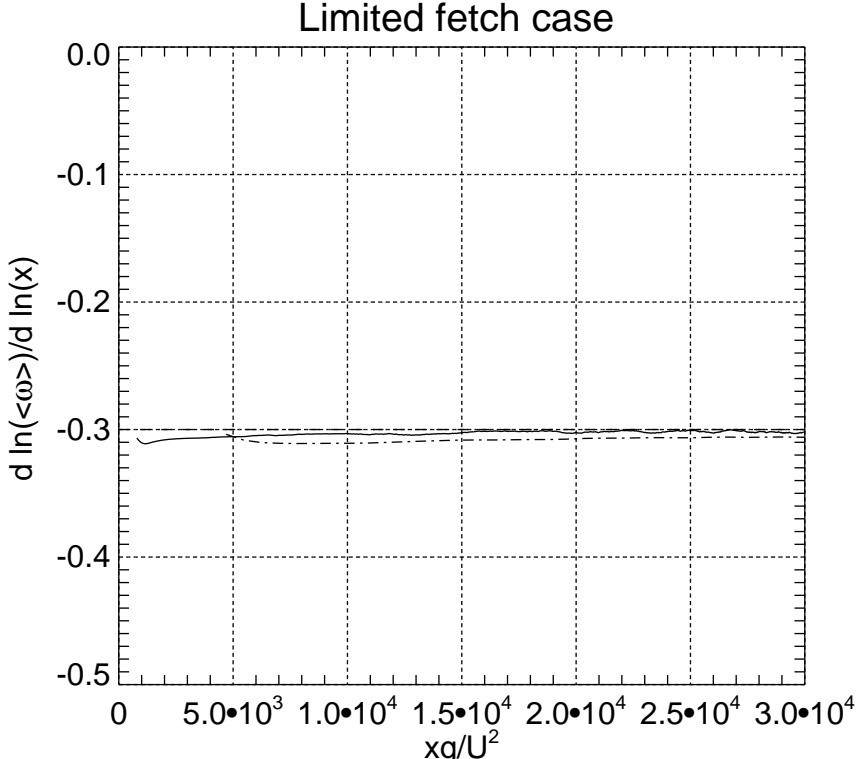

**Figure 15.** Local mean frequency exponent $-q = \frac{d\ln<\omega>}{d\ln x}$ as the function of dimensionless fetch $xg/U^2$ for fetch limited case. Wind speed $U = 10$ m/sec - solid line, wind speed $U = 5$ m/sec - dashed line. Horizontal dashed line - target value of the self-similar exponent $q = 0.3$.

Galtier, S. and Nazarenko, S.: Turbulence of Weak Gravitational Waves in the Early Universe, https://arxiv.org/abs/1703.09069v2, 2017.

Galtier, S., Nazarenko, S., Newell, A., and Pouquet, A.: A weak turbulence theory for incompressible magnetohydrodynamics, Journal of Plasma Physics, 63, 447 – 488, 2000.

Golitsyn, G.: The Energy Cycle of Wind Waves on the Sea Surface, Izvestiya, Atmospheric and Oceanic Physics, 46, 6–13, 2010.

Hasselmann, K.: On the non-linear energy transfer in a gravity-wave spectrum. Part 1. General theory, Journal of Fluid Mechanics, 12, 481 – 500, 1962.

Hasselmann, K.: On the non-linear energy transfer in a gravity wave spectrum. Part 2. Conservation theorems; wave-particle analogy; irrevesibility, Journal of Fluid Mechanics, 15, 273 – 281, 1963.

Irisov, V. and Voronovich, A.: Numerical Simulation of Wave Breaking, Journal of Physical Oceanography, 41, 346 – 364, 2011.

Janssen, P.: The Interaction of Ocean Waves and Wind, Cambridge monographs on mechanics and applied mathematics, Cambridge U.P., 2009.

Korotkevich, A. O., Pushkarev, A. N., Resio, D., and Zakharov, V. E.: Numerical verification of the weak turbulent model for swell evolution, Eur. J. Mech. B - Fluids, 27, 361 – 387, 2008.

Long, C. and Resio, D.: Wind wave spectral observations in Currituck Sound, North Carolina, JGR, 112, C05 001, 2007.

Longuet-Higgins, M. S.: A technique for time-dependent, free-surface flow, Proc.R.Soc.Lond. A, 371, 441 – 451, 1980a.

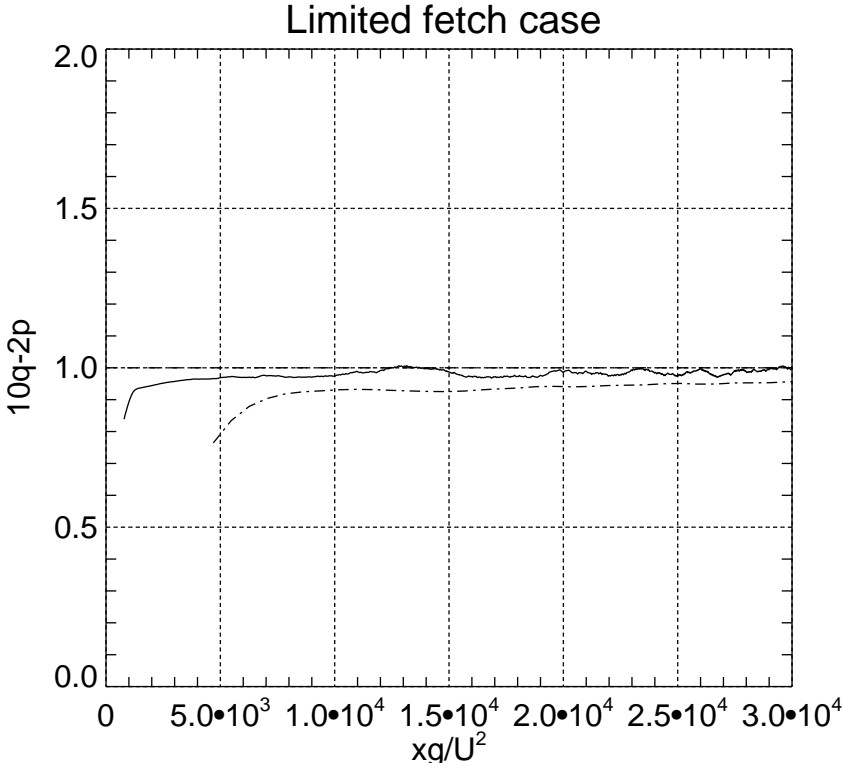

**Figure 16.** "Magic number" $10q - 2p$ as a function of dimensionless fetch $xg/U^2$ for fetch limited case. Wind speed $U = 10$ m/sec - solid line, wind speed $U = 5$ m/sec - dashed line. Horizontal dashed line - self-similar target value $10q - 2p = 1$.

Longuet-Higgins, M. S.: On the forming of sharp corners at a free surface, Proc.R.Soc.Lond. A, 371, 453 – 478, 1980b.

L'vov, V. S. and Nazarenko, S.: Spectrum of Kelvin-wave turbulence in superfluids, JETP Letters, 91, 428 – 434, 2010.

Nordheim, L. W.: On the Kinetic Method in the New Statistics and Its Application in the Electron Theory of Conductivity, Proc. R. Soc. Lond. A, 119, 689 – 698, 1928.

5  Perrie, W. and Zakharov, V. E.: The equilibrium range cascades of wind-generated waves, Eur. J. Mech. B/Fluids, 18, 365 – 371, 1999.

Phillips, O. M.: The dynamics of the upper ocean, Cambridge monographs on mechanics and applied mathematics, Cambridge U.P., 1966.

Phillips, O. M.: Spectral and statistical properties of the equilibrium range in wind-generated gravity waves, Journal of Fluid Mechanics, pp. 505 – 531, 1985.

Pushkarev, A. and Zakharov, V.: Limited fetch revisited: comparison of wind input terms, in surface wave modeling, Ocean Modeling, 103,

10   18 — 37, doi:10.1016/j.ocemod.2016.03.005, 2016.

Pushkarev, A., Resio, D., and Zakharov, V.: Weak turbulent approach to the wind-generated gravity sea waves, Physica D, 184, 29 – 63, 2003.

Pushkarev, A. N. and Zakharov, V. E.: Turbulence of Capillary Waves, Phys. Rev. Lett., 76, 3320, 1996.

Resio, D. and Perrie, W.: Implications of an $f^{-4}$ equilibrium range for wind-generated waves, JPO, 19, 193 – 204, 1989.

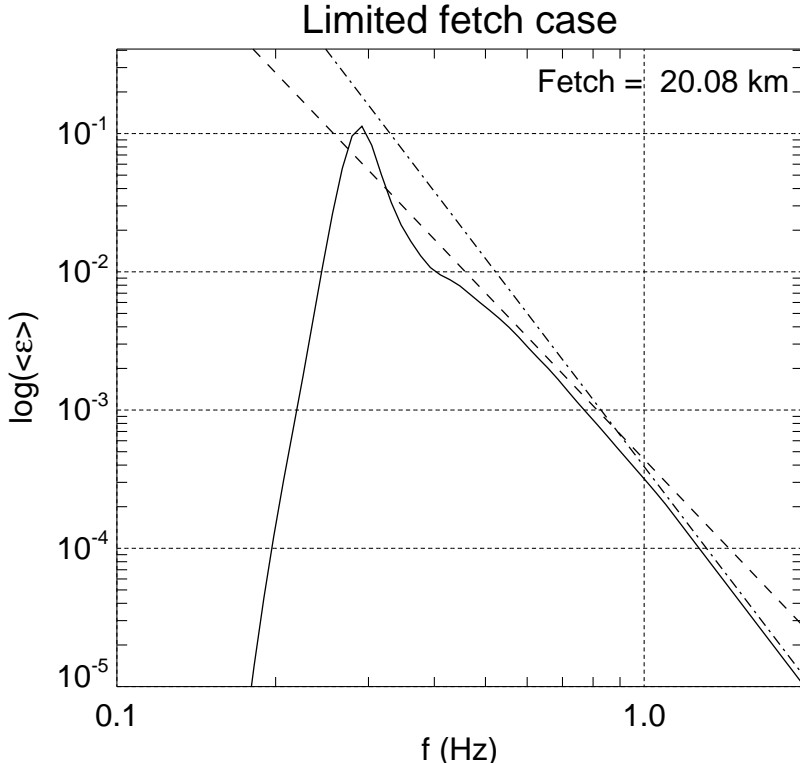

**Figure 17.** Decimal logarithm of the angle averaged spectrum as the function of the decimal logarithm of the frequency for wind speed $U = 10$ m/sec duration limited case- solid line. Spectrum $\sim f^{-4}$ - dashed line, spectrum $\sim f^{-5}$ - dash-dotted line.

Resio, D., Long, C., and Perrie, W.: The Role of Nonlinear Momentum Fluxes on the Evolution of Directional Wind-Wave Spectra, Journal of Physical Oceanography, 41, 781 – 801, 2011.

Resio, D. T. and Long, C. E.: Wind wave spectral observations in Currituck Sound, North Carolina, J. Geophys. Res., 112, C05 001, 2007.

Resio, D. T. and Perrie, W.: A numerical study of nonlinear energy fluxes due to wave-wave interactions in a wave spectrum. Part I: Methodology and basic results, J. Fluid Mech., 223, 603 – 629, 1991.

Resio, D. T., Long, C. E., and Vincent, C. L.: Equilibrium-range constant in wind-generated wave spectra, J. Geophys. Res., 109, C01 018, 2004a.

Resio, D. T., Long, C. E., and Vincent, C. L.: Equilibrium-range constant in wind-generated wave spectra, J. Geophys. Res., 109, CO1018, 2004b.

SWAN: http://swanmodel.sourceforge.net/, 2015.

Thomson, J., D'Asaro, E. A., Cronin, M. F., Rogers, W. E., Harcourt, R. R., and Shcherbina, A.: Waves and the equilibrium range at Ocean Weather Station P, JGR, 118, 1–12, 2013.

Tolman, H. L.: User manual and system documentation of WAVEWATCH III, Environmental Modeling Center, Marine Modeling and Analysis Branch, 2013.

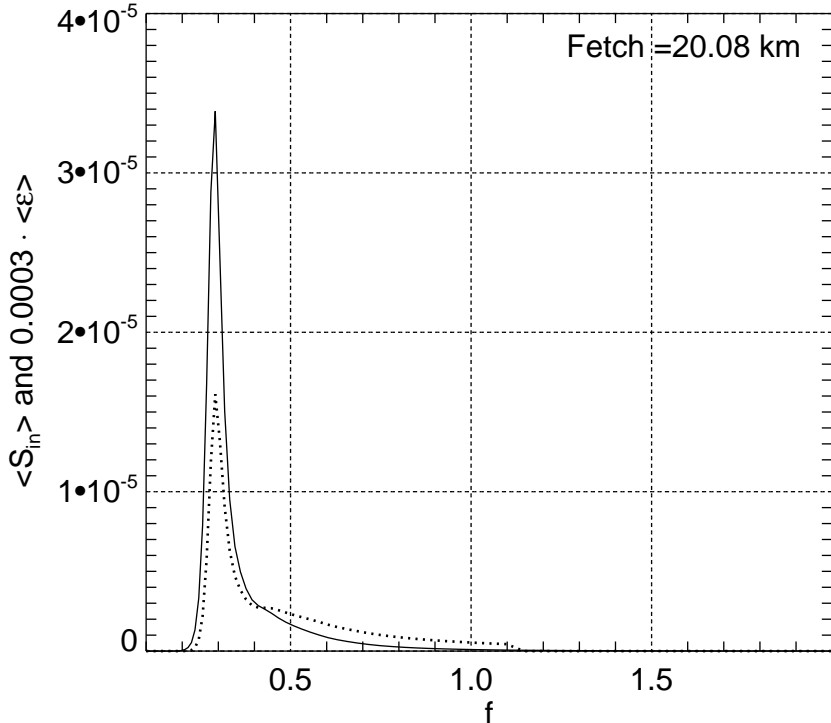

**Figure 18.** Typical, angle averaged, wind input function density $< S_{in} >= \frac{1}{2\pi} \int \gamma(\omega,\theta)\varepsilon(\omega,\theta)d\theta$ (dotted line) and angle averaged spectrum $< \varepsilon >= \frac{1}{2\pi} \int \varepsilon(\omega,\theta)d\theta$ (solid line) as the functions of the frequency $f = \frac{\omega}{2\pi}$ for wind speed $U = 10$ m/sec fetch limited case.

Tracy, B. and Resio, D.: Theory and calculation of the nonlinear energy transfer between sea waves in deep water, WES report 11, U.S. Army Engineer Waterways Experiment Station, Vicksburg, MS, 1982.

Tran, M. B.: On a quantum Boltzmann type equation in Zakharov's wave turbulence theory, https://nttoan81.wordpress.com/, 2017.

Webb, D. J.: Non-linear transfers between sea waves, Deep-Sea Res., 25, 279 – 298, 1978.

Yoon, P. H., Ziebell, L. F., Kontar, E. P., and Schlickeiser, R.: Weak turbulence theory for collisional plasmas, Physical Review E, 93, 033 203, 2016.

Young, I. R.: Wind Generated Ocean Waves, Elsevier, 1999.

Yousefi, M. I.: The Kolmogorov-Zakharov Model for Optical Fiber Communication, IEEE Transactions on Information Theory, 63, 377 – 391, 2017.

Yulin, P.: Understanding of weak turbulence of capillary waves, http://hdl.handle.net/1721.1/108837, 2017.

Zakharov, V. E.: Theoretical interpretation of fetch-limited wind-driven sea observations, NPG, 13, 1 – 16, 2005.

Zakharov, V. E.: Energy balances in a wind-driven sea, Physica Scripta, T142, 014 052, 2010.

Zakharov, V. E. and Badulin, S. I.: On energy balance in wind-driven sea, Doklady Akademii Nauk, 440, 691 – 695, 2011.

Zakharov, V. E. and Badulin, S. I.: The generalized Phillips' spectra and new dissipation function for wind-driven seas,
http://arxiv.org/abs/arXiv:1212.0963v2, 2012.

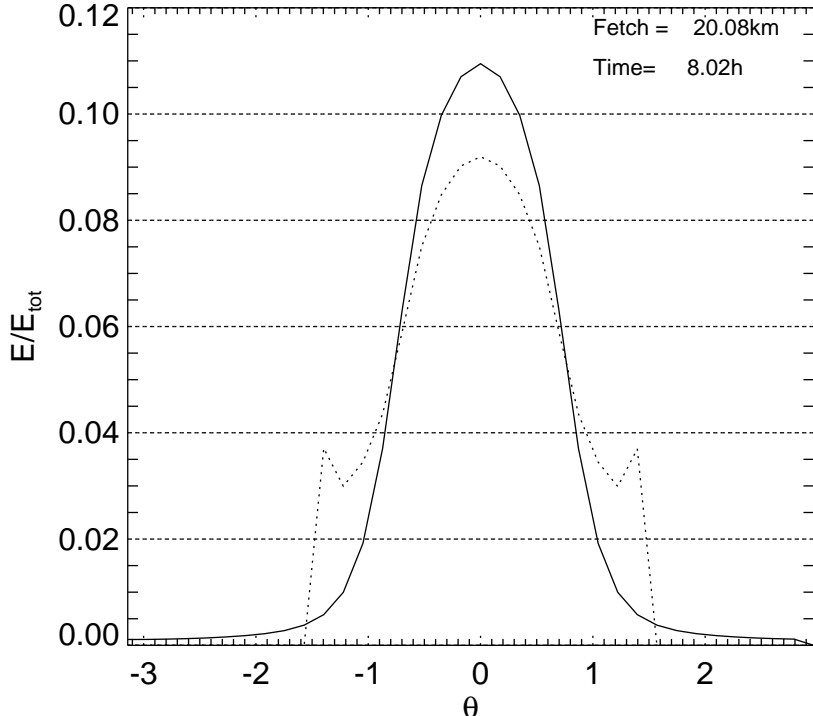

**Figure 19.** Relative wave energy distribution $E(\theta)/E_{tot} = \int_{f_{min}}^{f_d} \varepsilon(\omega,\theta)d\omega / \int_{f_{min}}^{f_d} \int_0^{2\pi} \varepsilon(\omega,\theta)d\omega d\theta$ as the function of angle $\theta$ for the duration (solid line) and limited fetch (dotted line) cases.

Zakharov, V. E. and Filonenko, N. N.: The energy spectrum for stochastic oscillation of a fluid's surface, Dokl.Akad.Nauk., 170, 1992 – 1995, 1966.

Zakharov, V. E. and Filonenko, N. N.: The energy spectrum for stochastic oscillations of a fluid surface, Sov. Phys. Docl., 11, 881 – 884, 1967.

5   Zakharov, V. E., L'vov, V. S., and Falkovich, G.: Kolmogorov Spectra of Turbulence I: Wave Turbulence, Springer-Verlag, 1992.

Zakharov, V. E., Korotkevich, A. O., and Prokofiev, A. O.: On Dissipation Function of Ocean Waves due to Whitecapping, in: American Institute of Physics Conference Series, edited by Simos, T. E., G.Psihoyios, and Tsitouras, C., vol. 1168, pp. 1229 – 1237, 2009.

Zakharov, V. E., Resio, D., and Pushkarev, A.: New wind input term consistent with experimental, theoretical and numerical considerations, http://arxiv.org/abs/1212.1069/, 2012.

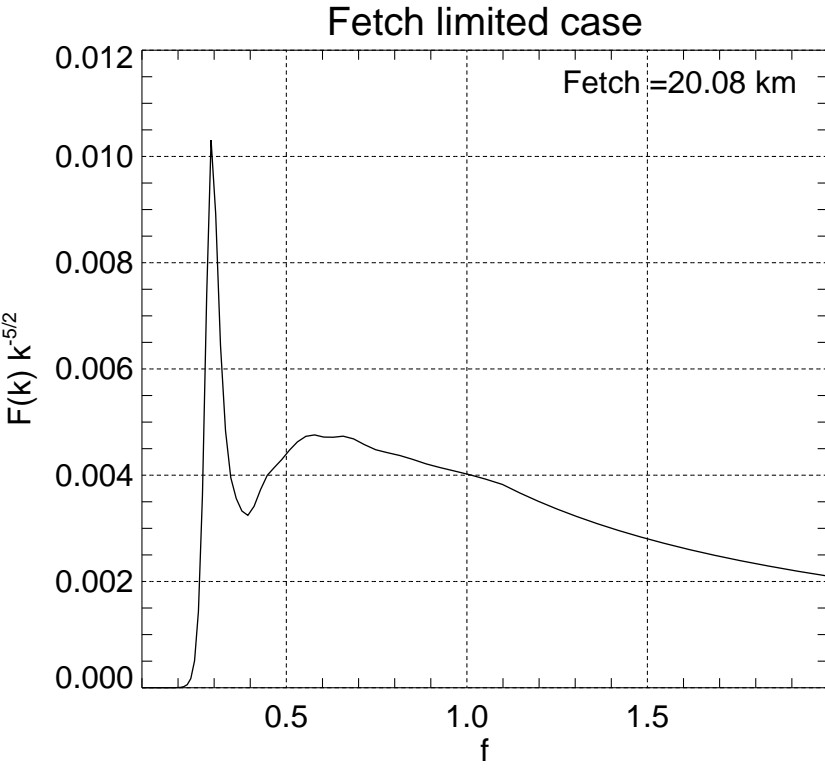

**Figure 20.** Compensated spectrum for fetch limited case as the function of linear frequency $f$ for wind speed 10 m/sec fetch limited case.

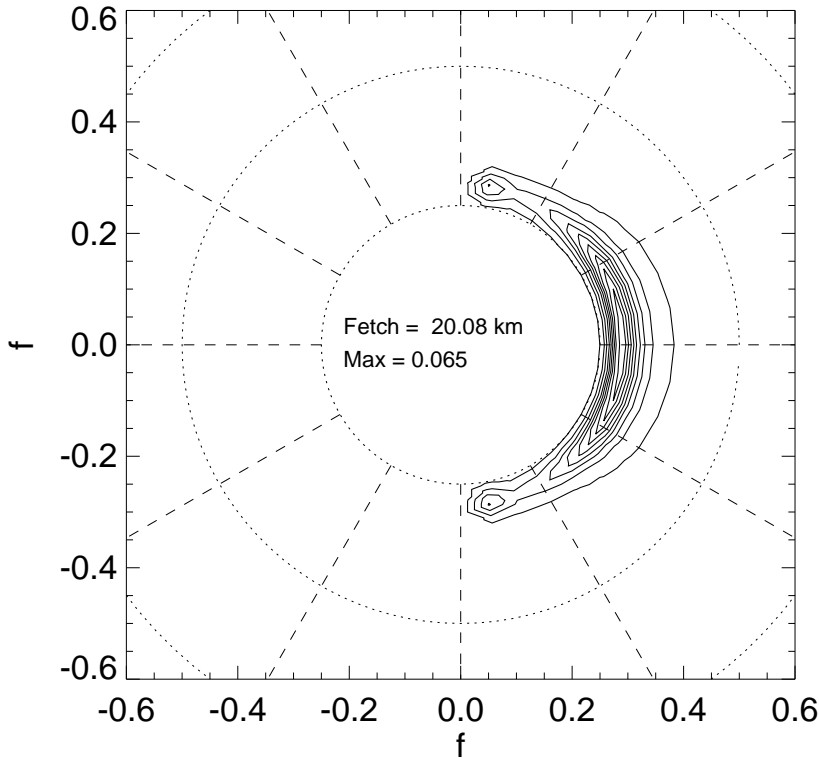

**Figure 21.** Angular spectra for wind speed $U = 10$ m/sec fetch limited case.

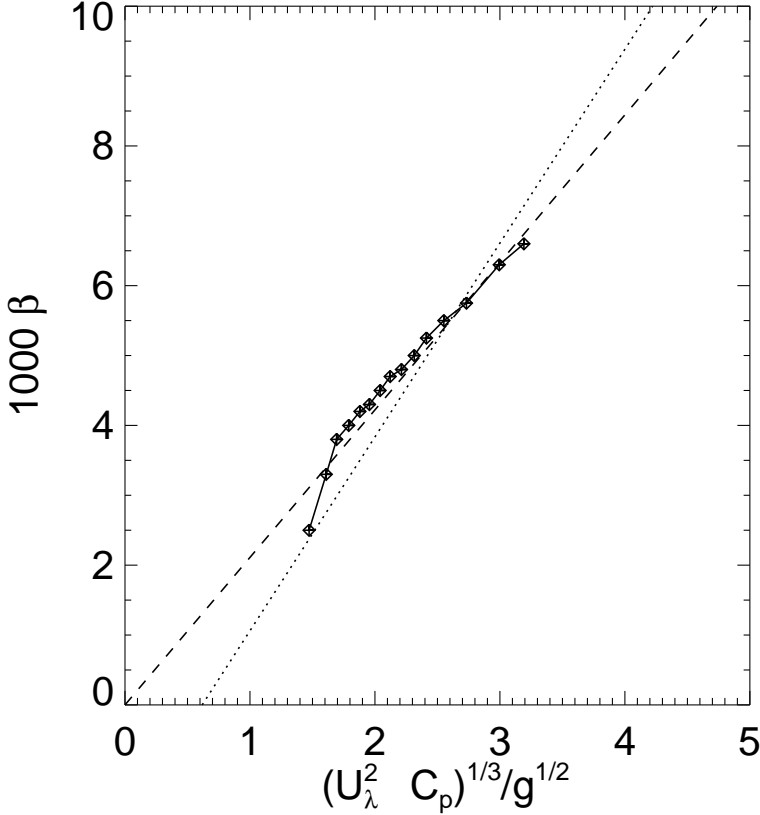

**Figure 22.** Experimental, theoretical and numerical evidence of the dependence of $1000\beta$ on $\left(u_\lambda^2 c_p\right)^{1/3}/g^{1/2}$. Dashed line - theoretical prediction Eq.(10) for $\lambda = 2.11$; dotted line - experimental regression line from Resio et al. (2004a); Resio and Long (2007). Line connected diamonds - the results of numerical calculations for wind speed $U = 10$ m/sec fetch limited case. Being parameterized by dimensionless fetch coordinate $\chi = \frac{xg}{U^2}$, the numerical simulation trajectory evolves from the left to the right on the graph, covering fetch span from $\chi = 0$. to $\chi \simeq 3.0 \cdot 10^4$.

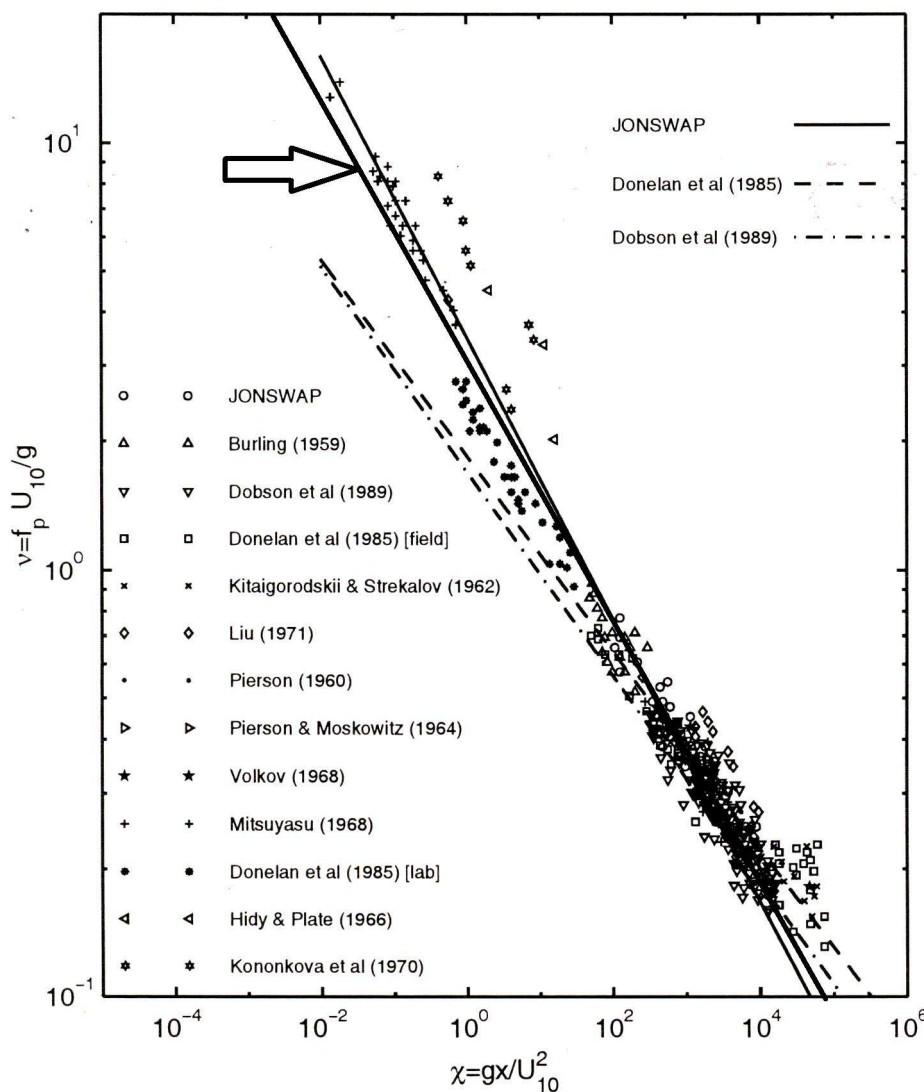

**Figure 23.** Solid line (pointed by arrow), presents non-dimensional energy $\varepsilon$ from the limited fetch numerical experiments, superimposed on the Figure 5.4, which is adapted from Young (1999). The original caption is: "A composite of data from variety of studies showing the development of the non-dimensional energy, $\varepsilon$ as a function of non-dimensional fetch, $\chi$. The original JONSWAP study (Hasselmann et al., 1973) used the data marked, JONSWAP, together with that of Burling (1959) and Mitsuyasu (1968). Also shown are a number of growth curves obtained from the various data sets. These include: JONSWAP Eq.(5.27), Donelan et al (1985) Eq.(5.33) and Dobson et al (1989) Eq.(5.38)."

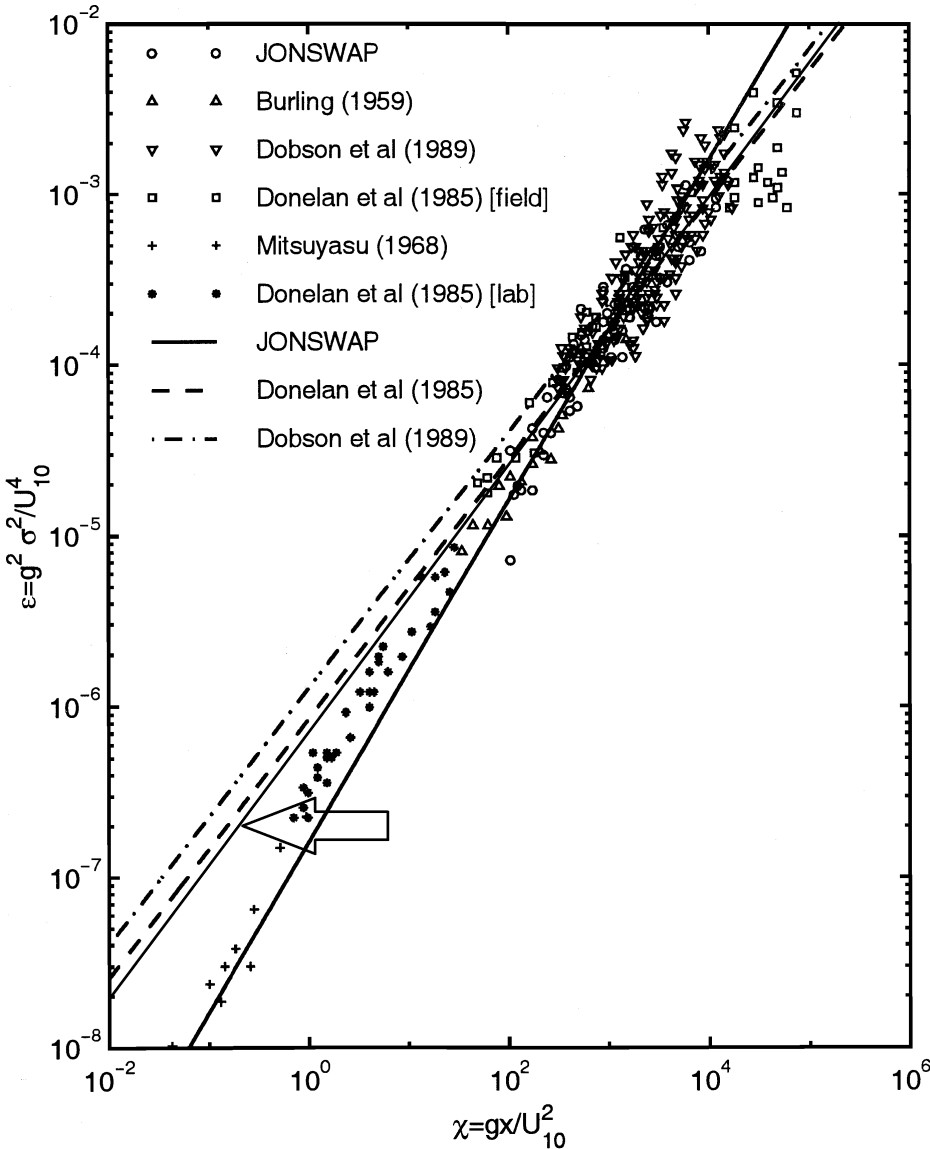

**Figure 24.** Solid line (pointed by arrow), presents non-dimensional frequency as the function of the fetch for limited fetch numerical experiments, superimposed on the Figure 5.5, adapted from Young (1999). The original caption is: "A composite of data from a variety of studies showing the development of the non-dimensional peak frequency, $\nu$ as a function of non-dimensional fetch, $\chi$. The original JONSWAP study (Hasselmann et al, 1973) used all the data shown with the exception of that marked Donelan et al (1985) and Dobson et al (1989). Also shown are a number of growth curves obtained from the various data sets. These include: JONSWAP Eq.(5.28), Donelan et al (1985) Eq.(5.34) and Dobson et al (1989) Eq.(5.39)."