# Peer review of "Balanced Source Terms for Wave Generation Within The Hasselmann Equation"

_Nonlinear Processes in Geophysics, 2016_

## Referee Comment (RC1) · Anonymous Referee #1 · 4 Jan 2017

The authors undertake a series of numerical experiments to investigate the form of the atmospheric input term required to reproduce observed cases for fetch and duration limited wave growth. The paper appears to be technically sound and will be of interest to the community. The quality of the paper would be enhanced by an edit from someone with English as their first language.

---

## Referee Comment (RC2) · Anonymous Referee #2 · 5 Jan 2017

This manuscript considers a new paradigm of the source term balance in deep water in spectral wave models. Key features are the role of non-linear four-wave interactions, a new concept of wave dissipation and a wind input closure term to arrive at self-similar solutions of the Hasselmann equations. This is interesting work deserving attention, but not in its present form. As outlined below that are too many issues that hamper publication. Therefore, I regret to advise a reject.

It seems that the present manuscript was put together much too hastily; insufficient attention has been paid to a proper description of the numerical results, there is hardly a comparison with observations, the representativeness of the chosen set of experiments is poor, and wishful thinking as expressed by a much too qualitative assessment of result.

[Figure]

Also, a large part of this manuscript is not new. The basic concept of a ZRP wind source term in relation to self-similar solutions has already been presented in Zakharov et al. (2012), and Pushkarev and Zakharov (2016, Ocean Modelling, 103). The fetch-limited tests have already been presented in PZ 2016. This applies to most figures related to fetch-limited wave growth. The only new results are related to checking the consistency of the new paradigm for duration limited wave growth. This in itself is too limited for publication in NPG.

It is noted that physical basis of the new ZRP wind input is missing. It is constructed as a closure term to enable self-similar solutions. Still, it is interesting that some features of wave growth are represented, of which the typical spectral shape deserves more attention.

There are hardly any comparisons against measurements, and the ones shown already appeared in PZ 2016. It is a shortcoming that no attempt has been done to compare the typical spectral shapes of Figure 7 and 17 with field observations.

The discussion of the results is poor, especially in section 4. Many figures are just mentioned with hardly any discussion. This also holds for the flow of the body text.

The number of numerical simulations is too limited to draw firm conclusions and the results shown are not convincing. Just consider Figure 11 where only 4 symbols should provide evidence of this set of source terms, or Figure 21 with only 7 symbols which do not even coincide with theoretical results. Details of the numerical procedure to handle the implicit damping are missing. Also section 4.1 in PZ2016 does not give implementation details. This makes it practically impossible to reproduce findings.

The range of applicability for other than academic 1D-cases is not discussed at all. As mentioned in PZ2016 the next step should be to test the applicability in 2D- field cases, but nothing is said about this.

The title is not appropriate. The focus is not on wind only, but on the whole concept of

the new set of source terms.

The quality of various fits or numerical results is poor, and the assessment of the goodness of fit is much too qualitative. An example is figure 2 where the fitted line is still far away from the solid line. Other examples are those in the figures 6 and 13, where the numerical results are still far away from the theoretical result.

The figures 10 and 20 hardly provide added value to the present study.

Many legends to figures are incomplete or cryptic.
* * *

---

## Author Comment (AC1) · 14 Mar 2017

Comment: The quality of the paper would be enhanced by an edit from someone with English as their first language.

Answer: The manuscript has been proofread by Dr. Donald Resio (co-author), native American speaker.
* * *

---

## Author Response (AR1)

**Answer to Referee #1**

**Question: The quality of the paper would be enhanced by an edit from someone with English as their first language.**

Answer: The manuscript has been proofread by co-author Dr.Donald Resio, native American speaker.

**Answers to Referee #2**

Question 1: Also, a large part of this manuscript is not new. The basic concept of a ZRP wind source term in relation to self-similar solutions has already been presented in Zakharov et al. (2012) …

**Answer 1: Derivation of new ZRP wind input term was first presented in V.Zakharov, D.Resio, A.Pushkarev, New wind input term consistent with experimental, theoretical and numerical considerations, 2012 arXiv:1212.1069, which is not peer-reviewed preprint. So, the current publication is the first presentation of the subject in academic stadarts media, essentially enhanced and upgraded.**
* * *
*Question 2:* The fetch limited tests have already been presented in PZ 2016. This applies to most figures related to fetch-limited wave growth.

**Answer 2: The fetch limited tests have been extended from 200 km to 300 km fetch, and all relevant figures have been upgraded.**
* * *
Question 3:The only new results are related to checking the consistency of the new paradigm for duration limited wave growth. This in itself is too limited for publication in NPG.

**Answer 3: The duration limited statement is as important as the limited fetch one. To the Authors opinion, studying of that situation itself in the context of self-similar properties of Hasselmann equation, would be sufficient for publication in NPG.**

**The Authors, however, went extra mile for presentation of general view on the wind-driven ocean waves development including not only duration limited, but also the limited fetch results in the context of analytic, experimental and numerical self-similarity aspects.**
* * *
Question 4: It is noted that physical basis of the new ZRP wind input is missing. It is constructed as a closure term to enable self-similar solutions.

*Answer 4: The physical basis of the new ZRP wind input term consists in the fact that it is the analytical self-similar solution of Hasselmann kinetic equation for waves, derived from Euler equation for free water surface.*

*The absence of the physical basis would mean that one or several following points are true:*

1. *Euler equations for free-surface flow doesn't have physical basis.*
2. *The physical basis was violated during Hasselmann equation derivation from Euler equations*
3. *The self-similar solutions are not the solutions of the Hasselmann equation*
4. *The physical basis has been lost during analysis of Resio and Long experiments 2004, 2007.*

*It is fare to ask the Reviewer to elaborate, at which stage the physical basis was lost.*

Question 5: There are hardly any comparisons against measurements, and the ones shown already appeared in PZ 2016. It is a shortcoming that no attempt has been done to compare the typical spectral shapes of Figure 7 and 17 with field observations.

*Answer 5: The universality of $\omega^{-4}$ asymptotics for large frequency is the worldwide recognized fact, observed in multiple experimental field observations, accepted by the oceanographic society after the seminal work of O.Phyllips, 1985. Citation has been added to the new version fo manuscript.*

*The extra Section 5 Comparison with the experiments has been added along with two new graphs comparing presented numerical results with the field experimental measurements.*
* * *
Question 6: The discussion of the results is poor, especially in section 4. Many figures are just mentioned with hardly any discussion. This also holds for the flow of the body text.

*Answer 6: The discussion of the results has been enhanced in connection with the relevant figures.*
* * *
Question 7: The number of numerical simulations is too limited to draw firm conclusions and the results shown are not convincing. Just consider Figure 11 where only 4 symbols should provide evidence of this set of source terms, or Figure 21 with only 7 symbols which do not even coincide with theoretical results.

***Answer 7: The number of points have been increased along with zooming out of the significant areas of the graphs.***
* * *
Question 10:  Details of the numerical procedure to handle the implicit damping are missing.

***Answer 10: The new ==Section 4 Numerical validation of relationship== have been added, which includes the subsection ==4.1 The details of "implicit" dissipation==***

---

## Author Response (AR2)

**Answers to Referee**

*The manuscript text, related to the first submission is printed in black, related to second submission - in red, related to the third submission – in blue.*

Question 1: The title is still not appropriate. The focus is not on wind only, but on the whole concept of the new set of source terms.

**Answer 1: The title has been changed to "Balanced Source Terms for Wave Generation Within Hasselmann Equation".**
* * *
Question 2: All computations have been performed with only one wind speed U=10 m/s. This may be OK to point out some seemingly interesting findings. But it lacks generality, especially in relation to the choice of a fixed frequency f=1.1 Hz where the Phillips tail is forced on the spectrum to mimic the implicit dissipation. Is the present method also applicable for low wind speeds in the order of a few m/s?

**Answer 2: To address the Referee concern, we numerically solved the same limited fetch Caughy problem for wind speed 5 m/sec. As a result, five graphs have been updated to accommodate the corresponding characteristics of the wave energy and frequency.**

**The comparison of the results, corresponding to wind speeds of 5 and 10 m/sec demonstrates that the both situations are consistent with self-similar solutions of Hasselmann equation with the tendency to asymptotically converge with growing fetch distance.**

**As seen from the graphs, the 10 m/sec case demonstrates better convergence with the fetch growth to theoretical self-similar prediction, than 5 m/sec one, which has the following explanation.**

**Hasselmann equation can be renormalized through the re-scaling of the fetch coordinate and spectral density to the universal form, where the wind speed dependence will be contained in front of the nonlinear interaction term. This means that, the stronger is the wind speed, the bigger is Snl term, and, as the result, the faster is convergence to asymptotic self-similar solution.**

**Overall, the spread between 10 m/sec and 5 m/sec cases does not exceed 10% difference for energy and 2% difference for mean frequency as the functions of the fetch.**

**Relevant corrections have been made to the manuscript text.**

Question 3: The authors comment on the physical basis of the ZRP wind input by providing 4 arguments. These are valid arguments indeed, but in my opinion they miss my key comment. The ZRP wind input was derived by virtue of the implicit damping forcing the tail to an f-5 shape In Eq. (11) only source terms for Snl4 and Sinp exist, whereas no Sds term is present. So, only the COMBINATION of ZRP and implicit damping enables the self-similar spectra, but no conclusion can be drawn about the validity (or physical basis) of the individual processes of dissipation and wind input. It is interesting that this notion is expressed by the authors on page 2, see line 28 ( .. our explanation is simple but has the same consequences), and line 33 (..and INDIRECTLY confirmed …). These statements cannot be considered as proof that each individual mechanism is based on first principles. i.e. a physical basis. Therefore my original questions remains: what is the physical basis of the individual source terms for wind input and dissipation?

*Answer 3:The question is closely related to Weak Turbulence Theory (WTT) and the kinetic equation for waves as its major tool.*

*The Hasselmann Equation  (HE) is the kinetic equation for surface ocean waves, besides another WTT applications, such as kinetic equations for plasma, spin and liquid helium waves. It is sad that the discussion of HE in the context of WTT has been neglected by major part of the oceanographic community for the lifespan of generations. This community accepts, nevertheless, HE as the basis of the operational wave forecasting models, therefore believing de-facto in WTT without understanding its ramifications.*

*The format of current communication is not to lecture WTT, it can be found in the book by Zakharov, Lvov and Falkovich (1992). We will only stop on relevant details of WTT, playng crucial role in the current manuscript.*

*To our understanding, the Referree questions can be re-formulated as follows:*

*1. "How one can get the wind input term from the HE, which doesn't care to contain any dissipation term, while dissipation and wind input processes are interconnected?"*

*2."How the self-similar solution and corresponding wind input term, obtained from dissipationless HE, can be in agreement with the implicit dissipation in the form of Phillips  $\omega^{-5}$ tail?"*

*Let's start with the first question.*

As per WTT, the four-wave nonlinear interaction generates direct energy cascade P from low to high wave-number, which is realized through the solution of HE in the form $\varepsilon \sim \dfrac{P^{1/3}}{\omega^4}$ . That effect was found theoretically by Zakharov, Filonenko (1968) as the solution of the equation $S_{nl}=0$ in infinite Fourier space, which formally does not contain any explicit dissipation term.

This situation is identical to what is realized in incompressible liquid turbulence for large Reynold numbers. Zakharov-Filonenko spectrum is equivalent to famous Kolmogorov spectrum, which transfers the energy from large to small scales. Energy dissipation is realized due to viscosity, but the viscosity coefficient, i.e. the dissipation details, are not included into Kolmogorov spectrum expression.

Therefore, solution of the equation $S_{nl}=0$ is called Kolmogorov-Zakharov (KZ) spectrum.

What is then the relation of the KZ spectrum to the reality?

This question splits, in fact, to two:

- why this solution in infinite space is related to the solution in finite space
- why dissipationless HE solution is close to solution of HE with dissipation

Let's try to address the first question. Consider the system in 2D wavenumber space $(k_x, k_y)$ . It is the infinite domain, having infinite capacity for energy storage, or infinitely effective energy sink at $|k| \to \infty$ , the "infinite phase volume" on the language of WTT. The direct energy cascade, propagating to the infinity, is getting absorbed by this sink just like there is the regular energy dissipation term somewhere at high wave-numbers.

The natural question arizes – what kind of relation this academical infinite phase volume sink has to the reality, because there is no such thing as infinitely small waves in nature? The answer is: the presence of the absorbtion at the finite high enough specific wave scale still preserves those KZ solutions, found from the equation $S_{nl}=0$ .

This statement is directly experimentally confirmed for ocean surface, capillary water and liquid helium waves. All those situations have radically different inertial range (the wave-numbers band between characteristic energy input and characteristic waves energy dissipation), but show KZ solutions. As far as concerns KZ spectrum, it was routinely observed in multiple experiments,

*including field ocean wave tanks ones. The results, published before 1985, are summarized by Phillips [16]. Thereafter, they were observed by Resio and Long [18], [19]. Complete survay of all $\omega^{-4}$ measurements requires separate comprehensive paper, which is in our plans for future.*

*The answer to the second question is the following. This approach, formally looking academic, finds its connnection with real life for the reason of dominant role of $S_{nl}$ term in HE, the related discussion and references can be found in PZ2016. That's why the first approximation to the real life solution can be found from the "dissipationless" HE.*

*Let's get back now to the question: "How the self-similar solution, obtained from HE equation and ZRP wind input term can be in agreement with the implicit dissipation in the form of Phillips $\omega^{-5}$ tail?"*

*The qualitative part, explaining correspondence of direct energy flux association with KZ solutions and its absorbtion mechanism is explained above. But it's intuitively obvious that such replacement of energy absorbtion at the infinity by finite wave-numbers dissipation can correctly reflect power behavior of the spectrum, but possibly give the wrong level of the spectral energy.*

*There is a lot of freedom in choosing the dissipation term. Since there is no current interpretation of the wave-breaking dissipation mechanism, one can choose it in whatever shape she/he likes, but any particular choice will be questioned since it is an artificial one.*

*Because of that, our motivation was that at the current stage, we need to know the effective sink with the simplest structure. If we continue the spectrum from some specific point with Phillips $A\omega^{-5}$ law, which decays faster than equlibrium $\omega^{-4}$ spectrum, we will get some unknown form of dissipation. We don't know the corresponding analytic parameterization of this dissipation term, while don't say that it's not possible to figure it out in some way. But we know that its exhibition is in the form of Phillips spectrum $\omega^{-5}$. One should note that this method of dissipation is not our invention. It was proposed by P.Jannsen and used in the WAM model, proper citation has been added.*

*The starting point of this "implicit" dissipation is still unknown. Now comes the experimental observation by Resio and Long, saying that that the transition from $\omega^{-4}$ to $\omega^{-5}$ spectrum happens approximately at the point $f_d = 1.1\,Hz$. We are "forcing" the continuation of the $\omega^{-4}$ spectrum at this point by $\omega^{-5}$. The spectrum amplitude at this junction frequency point is dynamically changing in time. What is important, is that this analytic continuation has inverse action to*

*the whole wave energy spectrum, since on every time step the*
*nonlinear interaction term $S_{nl}$ is calculated over the whole*
*"dynamic" and Phillips areas. Therefore, the Phillips part of the*
*spectrum "sends" the information about presence of the dissipation*
*above $f=1.1\,Hz$ to the other parts of the spectrum.*

*The whole set of the input and dissipation terms is accomplished now*
*with one uncertainty: the approach explained in the manuscript leaves*
*one parameter arbitrary – the constant in front of the wind input*
*term. We choose it from the condition of the reproduction of the*
*field observations wave energy growth as equal to 0.05.*
* * *
Question 4: The numerical implementation of the implicit damping in
the form of a Phillips f-5 tail has still not properly been
explained. Just noting that details are of secondary importance
cannot be an excuse. A key requirement of any paper is
reproducibility. Further, the choice of applying such a tail always
from f=1.1 Hz may be valid in the range of wind speed observed in
Resio and Long (2007) and here for U10=10 m/s, but it may fail for
lower wind speeds. In the extreme of a wind speed of 1.15 m/s the
Pierson-Moskowitz peak frequency coincides with 1.1 Hz. This
limitation and the applicability for low wind speed should be
discussed.

*Answer 4: The all details of numerical implementation of the implicit*
*damping in the form of $\omega^{-5}$ tails are included in the section 4.1 as*
*per Referee request in the first review. We have nothing more to add*
*to it. There is no any "criptic" content in it.*

*The Referee might be confused by the sentence "...the question of the*
*finer details of the high-frequency "implicit" damping structure is*
*of secondary importance, at the current "proof of the concept"*
*stage". It just means that current approach to the "implicit"*
*damping, starting at the fixed frequency $f=1.1\,Hz$ is obviously not*
*universal one, but rather crude first approximation, which might be*
*missing some finer detais of possible future, more sophisticated*
*approach to the $S_{diss}$ term. But even such simple approach is able to*
*produce consistent results. It is that simple.*

*As far as concerns Pierson-Moskowitz spectrum, its peak frequency is*
*given by $\dfrac{g}{2\pi U}$ . The transition to an $f^{-5}$ spectrum eventually can be*
*a dynamic variable as seen in the data of Long and Resio, but for now*
*we just ensure that it is sufficiently high, that the transition does*
*not feed back into the equilibrium range and peak-region solution.*
*The frequency 1.1 Hz is the peak frequency for a wind speed of about*
*1 m/sec, which was appropriate for the simulations in our paper.*

*Thus, this limit does not pose a problem for general applications.*
* * *
Question 5: On page 7, line 13 information is missing on the
frequency range and spacing.

*Answer 5: Corrected.*
* * *
Question 6: In addition, applying the implicit damping and forcing
the spectral shape for frequencies close to the peak frequency,
degrades the wave model concept from a 3G-model to a 2.5 G model.
This issue should also be discussed. It may limit the general
applicability of this method in wave forecasting techniques

*Answer 6: The assumption of a boundary condition at high frequencies
(i.e. a transition to a different form) in no way degrades the
solution.  The Boltzmann integral becomes unstable at very high
frequencies due to the fluxes coming into this zone.  This boundary
condition allows the fluxes to come into this zone, but does not
significantly affect the solution upstream from the flux boundary.
This is a totally appropriate boundary condition for this type of
application.  The boundary condition in existing 3G models is
inferior to this, since they simply "turn the interactions off' at
high frequencies.*
* * *
Question 7: Page 8, line 15. Details of the numerical procedure are
still missing. That details might be provided in a further paper is
no excuse to omit them here. It can't be that difficult to describe
these in a kind of pseudo-code which steps are taken in the numerical
procedure.

*Answer 7: The answer has been given in Answer 4, see also relevant
Answer 3.*

*Again, the line 15 of the Page 8 simply states that there is more
advanced analytical approach (Badulin and Zakharov 2012) to the
formulation of the dissipation term (not necessary implicit), than
used simple continuation of the spectrum by Phillips tail, but we
don't need it to formulate current manuscript statements.*

*To avoid the confusion caused by this statement, the reference has
been removed.*

Question 8: Is a constant time step used or does it depend on
dimensionless fetch or duration (as in the EXACT-NL model of

Hasselmann and Hasselmann, 1981).

**Answer 8: The constant time steps between 1 and 2 sec have been used in calculations to check the independence of the results on time discretness. Relevant comments have been added to the text.**
* * *
Question 9: Is the time stepping explicitly, or implicitly, etc…

**Answer 9: The time stepping is made through first order explicit scheme. Relevant comments have been added to the text.**
* * *
Question 10: Of particular interest is the treatment of the source terms in the action balance equation. It is therefore surprising that Eq. (45) contains a Sds term, whereas this term is missing in Eq. (11). These apparent inconsistency should be removed or at least explained if they are required for this manuscript. Does it hint that implicit damping is formulated in terms of a source term? If true, then the formulation of this source term should be provided.

**Answer 10: That is explained in Answer 4.**

**Again, the WTT assumes presence of perfect sink at infinite wave-numbers due to infinite phase volume, and therefore Eq.(11) doesn't incorporate the dissipation term. At this stage, the role of dissipation is played by infinite phase volume.**

**This WTT technique gives multi-parameter self-similar solutions family of HE.**

**Next step is the choice of the single parameter solution based on using of Resio-Long regression line. Now we have HE with ZRP wind input term, but unknown coeficint in front of it and absent dissipation term.**

**The next step is independent of the previous ones and consists in adding the "implicit" dissipation term, which will help us to deal with finite phase volume – there is no inifinite phase volume in numerical simulation in the reality, right?**

**Now we are almost done with HE model suitable for numerical simulation. The only missed thing is the coefficient in front of the wind input term, it is unknown. If we carry some numerical simulation with some arbitrary chosen coefficient, we will get many right qualitative properties of HE, like $\omega^{-4}$ spectrum, spectral peak down-shift and peak frequency behavior in accordance with self-similar laws, but wrong level of spectral energy.**

*How can we handle that? That coefficient has to be chosen to get the same wave energy growth as was observed in field experiments.*

*At this step we are done with the construction of the HE model, Eq. (45).*
* * *
Question 11:  On page 9, line 2 the universality of the omega-4 for large frequencies is mentioned. This statement needs clarification as it is not clear what is meant with LARGE frequencies. Are these higher than 1.1 Hz? Looking at Figure 7 a typical spectral shape is seen with an f-4 region just above the peak frequency and a Phillips tail for larger frequencies.

*Answer 11: The spectrum $\omega^{-4}$ , according to WWT, is realized for so-called "intertial" range, also known in oceanography as "equilibrium" range. Roughly speaking, it is the range between wave energy source (associated with the spectral energy peak) and beginning of the dissipation (frequency 1.1 Hz). One can't expect exact points of the beginning and ending of this spectrum, since they are smeared out due to transitions to the regions of energy input and dissipation.*

*The corresponding correction has been made.*
* * *
Question 12: Page 9, line 10. Which RMS value is referred too?

*Answer 12: RMS is drawn on Fig.1, adapted from Resio and Long 2007 as $r^2 = 0.939$ with corresponding reference.*
* * *
Question 13: Page 8, section 4.2. Which method was used to estimate the parameters p and q?

*Answer 13: As marked on the vertical axis, $p = \dfrac{d\ln(E)}{d\ln(t)}$ . First, we smooth the function E via moving average, then calculate the derivative numerically through taking finite differences, and subsequently moving averaging the result. The parameters of moving averages have to be found individually in any specific case.*

*The parameter q is calculated similarly.*

*Corresponding corrections have been made in the manuscript text.*
* * *
Question 14: Page 12, line 10. Please correct. It is the dimensionless total energy!

*Answer 14: Corrected.*

Question 15: Page 12, line 10. Please correct. It is the dimensionless fetch!

*Answer 15: Corrected*

Question 16: Page 12, line 12. Please correct. It is the dimensionless mean frequency!

*Answer 16: Corrected*

Question 17: The conclusions are a bit short. There is hardly a serious discussion on the application of this promising method for other cases including applications to lower wind speeds.

*Answer 17: There is no problem with low wind speeds if we set the transition frequency to 2.0 hz.*

Question 18: The range of applicability for other than academic 1D-cases is not discussed at all. As mentioned in PZ2016 the next step should be to test the applicability in 2D- field cases, but nothing is said about this. Neither about the applicability of implicit damping for low wind speeds when the value of 1.1 Hz may not be appropriate any more.

*Answer 18: The transition to 2D case requires radical increase of the calculations volume. Currently, we learned to use parallel computations and intended to sharply accellerate the computations. That is the way to perform 2D simulations.*

*The question about applicability of the implicit damping for low wind speeds has been addressed in previous answers.*

Question 19: The introduction and discussion of the results in the many figures presented is still poor. Some comments to a specific figure may also hold for other figures. My comments serve as a guideline for clarifying many detailed issues and the authors should check this carefully!

*Answer 19: The authors checked the figures, corrected the figures captions and improved relevant discussion.*
* * *
Question 20: Figure 2 contains a systematic deviation between results and the fit. This does not look like a fit as the lines only coincide at the origin. This difference is puzzling. Some lines are not explained in the legend to the figure or in body text.

*Answer 20: The relatively small systematic deviation of the fit is connected with two facts.*

*The first fact is the transition process in the beginning of the simulation, when the wave system behavior is far from self-similar one. But the fit is build as power function, corresponding to self-similar solution, without taking into account the initial transition process, and that causes the systematic difference. This systematic difference could be diminished via parallel shift of the fit, which would take into account the initial transition process – that is just equivalent to starting the simulation at different time.*

*The second fact is the asymptotic nature of the self-similar solution, and it's quite natural to observe the convergence of the simulated wave system toward self-similar behavior with the fetch coordinate growing, as seen on Fig.3.*

*In Authors opinon, the first fact is the major reason of systematic deviation.*

*The lines description is corrected.*
* * *
Question 21: Figure 3. Add the target value of p=10/7 and comment that the relative error is still 6%, which is clearly subjectively acceptable by the authors. It could also be noted that there is an asymptotic behavior for long duration. Whether this also occurs in nature, where conditions are less ideal, should be a point of

discussion.

*Answer 21: The target value p=10/7 has been added to the figure. The comment on asymtotic behavior of the self-similar solution with relative error of approximately 6% deviation from the target value p=10/7 for long duration added to the text.*

*As far as concerns what will happen when the conditions are less ideal. In nature, the varying forces will produce variations in the evolution of the wave spectrum, but it is critical to have the fundamental relationships to use as a basis for this evolution. which can be called "theoretical relationship".*
* * *
Question 22: The range of applicability for other than academic 1D-cases is not discussed at all. As mentioned in PZ2016 the next step should be to test the applicability in 2D- field cases, but nothing is said about this. Neither about the applicability of implicit damping for low wind speeds when the value of 1.1 Hz may not be appropriate any more.

*Answer 22: As noted previously, the transition frequency to the "flux dissipation zone" can easily be varied, without changing any of the practical implications of the methodology.  It is clear that this approach would need more work to be extended to a generalized model, we believe that this would be a significant step forward in the basic physics of the models.*

*The typical spectral shape is consistent with several previous publications such as Resio et al. (2004) and Long and Resio (2008). The directional behavior is consistent with observations as noted in Resio et al. (2011).*
* * *
Question 23: Figure 4: the different lines in the figure are not explained in the legend and in the body text. It is puzzling why the fitted line has a systematic difference with the computed line. I wonder whether it is a fit at all. In case it is a fit, then the method how the fit was made should be explained.

*Answer 23: The lines explanation has been added to the legend.*

*The dashed line is the self-similar solution with the empirically adjusted coefficient. The word "fit" has been removed from the text.*

*The reasons of the systematic  difference are the same as for Fig.2 - the transition process in the beginning of the simulation and*

*asymptotic nature of self-similar solution.*

*The relevant comments have been added to the text.*
* * *
Question 24: Fig 5: the magic target value of q is missing in this figure and legend. The occurrence of the wiggles is not noted and discussed. Further, there is a discrepancy between the sign of q in the body text and in the figure

*Answer 24: The target value of q has been added to Fig.5 as well in the legend.*

*The wiggles could be caused by finite number of resonant quadruplets used in the numerical simulation.*

*Discrepancy between the sign of q in the body text and in the figure has been corrected.*
* * *
Question 25: Figure 6: the magic target of p=1 is not plotted. The choice of the range along the vertical axis obscures the relative error of 10%, which is seemingly acceptable by the authors. Nowhere in the manuscript such differences are explained. Only subjectsive statements about 'goodness of fit' are made.

*Answer 25: The magic target p=1 has been plotted.*

*The vertical axis range has been changed for better seeing the relative error of 10%.*

*The nature of this difference has been explained in the manuscript text.*
* * *
Question 26: Figure 7: the dashed lines are not explained in the legend. The legend along the x-axis is wrong. Although the frequencies are plotted on a log-scale, the actual frequencies are shown. So the legend should just be f (Hz). Note that the unit should be added. Further no comments are made on the regions in the spectrum where the spectrum adheres to either an f-4 of f-5 tail. Some guidance to actual values enhances the readability.

*Answer 26: The dashed lines are explained. The legend along the x-*

*axis is corrected. The x-axis legend is changed to f(Hz). The comments on the regions of the spectum close to $f^{-4}$ and $f^{-5}$ are made.*
* * *
Question 27: This typical spectral shape is worth mentioning, especially as this shape has been observed in nature. It is unclear why the authors have not been searching for empirical evidence of this behavior. It could only strengthen their case and is one of the interesting results of this study.

*Answer 27: The same spectral shape has been observed by Resio and Long (2008).*

*Relevant comment has been added.*
* * *
Question 28: Figure 8 appears after figure 9. This should be reversed. The dashed line is not explained in the legend. The inertial range should be better explained as not all readers immediately see where to look.

*Answer 28: The figures order is reversed, the dashed line is explained in the legend.*

*The explanation of the inertial range has been added to the text.*
* * *
Question 29: Figure 10. The broadening of the spectrum is not visible in this type figure. A direct way, and more convincing, is to plot the directional spreading as a function of frequency. Take care of dimensions along figure axes.

*Answer 29: The Fig.10 and relevant comment should be better probably removed from the manuscript as not important to the general context of what we are saying.*
* * *
Question 30: Figure 11: The line types are not defined. The RMS error should be quantified. The range of values to which the dots refer should be mentioned as it is not yet stated which part of the simulation is covered. The convergence to theoretical results should be mentioned. Also, an explanation why the regression line has a systematic deviation to the computational results should be discussed. Usually, a fitted line has a certain minimum error and is

close to the data points, but not here. Details about the fit procedure should be provided, at least for reproducibility.

*Answer 30: The line types have been defined.*

*The RMS value from Fig.1 has been added in figure captions.*

*The comment regarding the range of the simulation the dots cover has been added.*

*The convergence to theoretical results was mentioned.*
* * *
Question 31: Figure 13: There is still an error of about 5% in the computed value of p. Wiggles appear in the simulation. This should be noted and explained in the context of this study. The choice of the range along the vertical scale subjectively improves the quality of the results. This may be OK, but only in combination with a quantitative assessment of the error.

*Answer 31: As per previous Referee comment, we re-calculated the wave system evolution for lower wind speed of 5 m/sec, and Fig.13 is updated now with the new result.*

*It was answered above that the wave evolution for wind speed 5 m/sec is expected to be slower than for 10 m/sec due to weaker nonlinear interaction term. We observe, indeed, slower convergence of the calculated exponent to target exponent value p=1 for 5 m/sec case, than for 10 m/sec case. The deviation of 10 m/sec case exponent from target value does not exceed the error of about 5%, while for 5 m/sec case doesn't exceed the error of about 20%. The role of relatively short in time non-self-similar evolution of the wave system at the very beginning should be noted as well as the factor contributing to the deviation from the target value of exponent p=1.*

*The small amplitude wiggles of the exponent evolution are attributed to the limited number of quadruplets used in the simulation.*
* * *
Question 32: Figure 14: The various lines in this figure are hardly explained in the body text. Further, fp is not mentioned. To what purpose have both fm and fp been plotted? There is no discussion about the relative position of these lines and whether this is acceptable. There seems to be a systematic bias in the results.

*Answer 32: Lines explanation has been added in the body text. Peak frequency line $f_p$ has been removed.*

*Small systematic bias has the same explanation as in previous*

*questions: the transition process in the beginning of the simulation and asymptotic nature of self-similar solution.*
* * *
Question 33: Figure 15, Text is missing about what can be seen in this figure. Also a mismatch in sign of q between body text and figure.

*Answer 33: The text has been added in the body of the manuscript.*

*The sign of q has been fixed.*
* * *
Question 34: Figure 16. Here is good agreement, but also some wiggles appear in the solution.

*Answer 34: The wiggles are apparently caused by limited number of resonant quadruplets used in the simulation.*
* * *
Question 35: Figure 17. The legend along the x-axis is wrong, it should be f (Hz) and probably also for the y-axis. The data are plotted on log-scale but the values remain unchanged. The dashed and dash-dot line are not explained in the legend.

*Answer 35: The x-axis legend has been fixed. The dashed and dash-dotted line have been explained.*
* * *
Question 36: Figure 18. The dotted line is not explained in the legend. An explanation is required why the ZRP wind-input term drops to zero for f> 1.1 Hz. This seems related to the numerical procedure, but has not been explained. It also contradicts the equations 41-44 where no mention was made of this behavior. It appears an essential part of the numerical procedure requiring explanation.

*Answer 36: The dotted line has been explained in the legend.*

*The angle-averaged line for the energy input, associated with with ZRP wind input term, drops to zero, indeed, for $f > 1.1\,Hz$. The frequency range of ZRP wind-input term definition has been added in Eq.(43): ZRP $S_{in}$ is growing with the frequency from $f_{min} = 0.1\,Hz$ to $f_d = 1.1\,Hz$. The wind input is equal zero thereafter, in the region of the "implicit" dissipation.*

Question 37: Figure 19. The units are missing along the x-axis. Also note the discontinuity of the compensated spectrum at f=1.1 Hz. This is a puzzling issue.

**Answer 37: The units have been added to x-axis. There is no discontinuity of the spectrum at $f=1.1\,Hz$, but rather the derivative discontinuity, the "kink", indicating the transition from $\omega^{-4}$ to $\omega^{-5}$ spectrum.**

Question 38: Figure 20. See also comments on Figure 10. Further, the origin of the secondary peaks at angels of +/-85° should be noted. Is this a serious side-effect?

**Answer 38: The equations solved for time domain and fetch limited situations are different, and, in our opinion, we shouldn't expect the angular distributions to be identical. The side peaks in angles are often observed in natural measurements.**

Question 39: Figure 21: Note the convergence to the theoretical results. Also, what range do the plotted symbols cover.

**Answer 39: Note on the convergence to the theoretical results has been added.**

**Being parameterized by dimensionless fetch coordinate $\chi = \frac{xg}{U^2}$, the numerical simulation trajectory evolve from the left to the right on the graph from $\chi = 0$ to $\chi = 3.0 \cdot 10^4$, the coresponding comment has been added.**

---

## Author Response (AR3)

**Answers to Referee:**

The manuscript text, related to the first submission is printed in black, to the second submission - in red, to the third submission – in blue, to the fourth submission – in lavender.
* * *
*It is a pity that the rebuttal gives additional clarifications which were not included in the modified body text. This is a missed chance of this revision.*

Answer: The clarifications from the third submission Answers to Referee have been assimilated into the enhanced Introduction.
* * *
*I read the rebuttal with interest which provides interesting additional information. Still my simple question is not answered. The wind input is evidently not derived from first principle but only made to make self-similar solutions possible. By key remark is that the wind input is not based on observations. This notion is now 'hidden' in the text and should be made more explicit. As it is still artificial, it would be of great interest when some validation against measurements is recommended.*

Answer: The explicit statement on ZRP wind input term is added to:

- Introduction, page 4, lines 3-8
- Conclusion, page 22, lines 3-6 and page 23, lines 1-2

The integral characteristics validations against measurements are provided in the section **8 Comparison with the experiments**.

While it is true that the details of the wind input term are not validated, the key point here is that the measured spectral shape is reproduced quite well, whereas, the current form of 3G source terms does not accomplish this. The shape of the spectrum provides a very good surrogate for understading the source term balance.
* * *
*Comment #4*
*The detailed description of the numerical scheme is now almost complete. The lines 15-16 on page 8 are a good starting point. It is still not clear when implicit damping is applied in the procedure. It may simply be resolved by adding something like:*
*1) Enew(f,theta) = Enew(f,theta)+dt\*[Swind(f,theta) + Snl4(f,theta)]*
*2) Overwrite Enew(f,theta) to f-5 tail for f>1.1 Hz*

*3) Compute Snl4 +Swind over full range of spectrum 4) Etc… Such an addition may enable reproducibility*

Answer:

The following pseudo-code has been added on page 11, line 32 and page 12, lines 1-4:

1. Calculate $S_{nl}(\varepsilon(f,\theta))$

2. Overwrite $\varepsilon(f,\theta)$ to $f^{-5}$ for $f > 1.1\,Hz$

3. Update $\varepsilon(f,\theta) = \varepsilon(f,\theta) + dt \cdot S_{nl}$

4. Solve analytically $\dfrac{\partial \varepsilon(f,\theta)}{\partial t} = S_{wind}$ for time $dt$

5. Return to step 1
* * *
**Comment #6**
*I disagree that the WAM approach for the tail is inferior to the present method. I note that both approaches force a parametric tail to the spectrum from some frequency. The similarity is that for this range the wind source terms do not play a role anymore, whereas both in the DIA and in the WRT the tail is used in the evaluation of the Snl4 term. These properties lead in my interpretation to a 2.5 G model. The differences, however, are related to the assumed physical origin.*

The treatment of the tail of the spectrum is required to maintain stability in the $S_{nl}$ integral. Otherwise, the fluxes run up against a "dam" and the energy levels become so large that they create instabilities in the integral. This allows the spectrum to maintain a classic $k^{-2.5}$ form in the equilibrium range.

We concur that both versions of the parametric tail are not in concurrence with a detailed-balance forms for a breaking source term as they are currently formulated.
* * *
**Comment #17**
**This section is still short. The rebuttal gives sufficient additional information to better elaborate on the position of this research in the long-term quest to a sound wave prediction model. To name a few: this study is a proof of concept. Which steps are needed to improve on this result, how to solve the issue of insufficient quadruplets, how to solve the strange blobs (Fig. 20), which may be solved by 2D-**

**wave model computations, validate ZRP wind input against measurements, validity for low and high wind speeds.**

The section **9 Conclusions** has been augmented to address Referee concerns on pages 22-24 on the following:

- the note on the study as the proof of the concept

- additional study of sufficient grid resolution which might be exhibited in oscillation of self-similar indices is required

- the spectral blobs appearance, corresponding to the waves running almost orthogonal to the fetch (well-known "smiley" effect), is presumably a numerical artifact connected with the specifics of the studied fetch limited statment. Despite it is shown that their relative contribution doesn't exceed $5\%$ of the total wave energy, and they are insignificant for the research purposes, their relevance to the reality can be studied via full 2D limited fetch simulation

- although the integral parameters of the model have been verified agains the experimental observations, more verification of the spectrum details, such as angular speading, is required

- the test of the model invariance with respect to wind speed change from 5 to 10 m/sec has already been done, but further study of the effects of wider range of wind speeds variation on self-similar properties of the model is desirable in the future
* * *
***Comment #18***
***These comments on the 2D application are worth including in the outlook for further work***

At the moment of submission of the manuscipt, the main techical obstacle to effective development of new generation of physically based HE models is insufficiently fast calulation of exact nonlinear interaction. The transition to 2D case requires radical increase of the calculations speed. We hope that further such improvements will be made in near future.

Relevant comments have been added to the section  8 Conclusions.
* * *
***Comment #22***
***Just add an explicit reference to Resio and Long (2007), which clearly illustrates the existence of a bump in the spectrum. It would be great when the authors may provide additional references.***

The corresponding references have been added in relevant places.
* * *
**Comment #24**
**The comments in the rebuttal about the origin of the wiggles (insufficient quadruplets) should be included in the body text. It is relevant information for a reader and also for reproducibility.**

The origin of the wiggles have been reformulated to more understandable language.

The number of quadruplets is connected with the grid resolution. The issue of limited quadruplets number is, in fact, the finite grid resolution issue, which exhibits itself, in particular, in the indices oscillations in the following way: the spectral peak is down-shifting in the process of the evolution, and most of the times its location is "in-between" neighboring grid point, while the self-similarity theory deals with continuous Fourier space. When the spectral peak coinsides with the grid node, its value jumps. That should be the reason of the observed indices oscillations.

We shifted from the quadruplet language explanation to grid resolution explanation (discretness) one, since it is more understandable to the reader.

Relevant comments have been added to the figures explanations as well as in the Conclusion.

**Comment #27**
**Just give explicit reference to Resio and Long (2007), and/or others.**

The corresponding references have been added in relevant places.
* * *
**Comment #29**
**I agree that removing this figure (Fig 10). It is not very illuminating and it is not a proper way to illustrate a directional broadening. Please take care in renumbering all remaining figures.**

We desided to leave the figures of frequncy-angular distribution and added the additional figure showing the portion of total energy, containing in every angle. This figure shows that the "smiley" effect, being presumably the numerical artifact, is not significant for our purposes.
* * *
**Comment #31**
*In the rebuttal a reasoning is given related to '… limited number of quadruplets …' I do not exactly understand what is meant which this phrase. I can only speculate that the present computations were carried out with a too coarse model to evaluate the nonlinear interactions. If true, it degrades the soundness of the present results. This issue should be clarified and shared with the reader.*

See the answers under **Comment #24**
* * *
**Comment #34 See previous comment**

See the answers under **Comment #24**
* * *
**Comment #36 The authors are a bit cheating here in explaining the fact that Swind=0 for f>1.1Hz. This behavior conflicts with the Eqs. 41-45, where a continuous function is presented without a frequency cut-off. So, wind input is artificially set to zero as part of the numerical procedure.**

The Authors don't see any cheating in here. Let us explain, why.

The wind input is not only artificially set to zero above f_d=1.1 Hz as the part of the numerical procedure – it is zero on the stage of the continuous model formulation.

The cheating, in our opinion, is in the construction of the WAM-like source terms, when the supposed-to-be physically based Snyder wind input term is superimposed with dubious spectral maximum dissipation function, producing as the result of their summation some sign-indefinite source function.

In our approach, every frequency range has to be occupied by inidividual physically based source term -- that's why the wind input stretches only up to f_d=1.1 Hz. As far as concerns discontinuity of the wind input function, it is chopped off on the stage of transition from WTT approach to the phase space confined model, corresponding to the reality (see the modified Introduction)

Now suppose the realization of the "non-cheating" case – that we desided to continue the wind input function up to the highest frequency. Anyhow, it has to be overlapped with some dissipation function at high frequencies, strong enough to suppress the wind input above f_d=1.1 Hz and stabilize the model from non-physical energy "build-up" at the highest frequency due to the "damb" effect (insufficient ability of the highest frequency bound to "leak" the KZ

energy flux), which would have some unpredictable reverberations at high frequencies, including energy flux reflection from frequency domain upper bound and numerical instabilities.

As far as concerns discontinuity of the source terms at f_d=1.1 Hz, it does not consitute any problem whatsoever, since the integral equations, in the contrary to the differential ones, exhibit solutions smootheness even for discontinuous source terms.
* * *
*Comment #38*
*The explanation offered by the authors to explain the spectral blobs at angles of +-85 degrees is in my opinion wrong. In addition, Figure 20 does not show directional broadening, unless the authors consider the blobs at either end of the spectrum as directional broadening. In case the authors stick to a physical explanation, then proper references should be given. In my opinion the blobs are spurious artefacts arising from the numerical procedure to evaluate fetch-limited wave growth using a simple 1d-wave model. For angles close to +-90 degrees the cos(theta) (see 46) is close to zero leading a strong growth of energy. This is similar to an infinite fetch perpendicular to the wind direction. This feature is known as the smiley effect for decades. Moreover, it manifests itself mainly close to shore and vanishes in 2D- wave model computations.*

We agree that this is the numerical artifact. See the answer to the **Comment 17 and 29**. Relevant explanation is also included at the section in the relevant figure comment and section **8 Conclusions**.